# MomentSeg: Moment-Centric Sampling for Enhanced Video Pixel Understanding

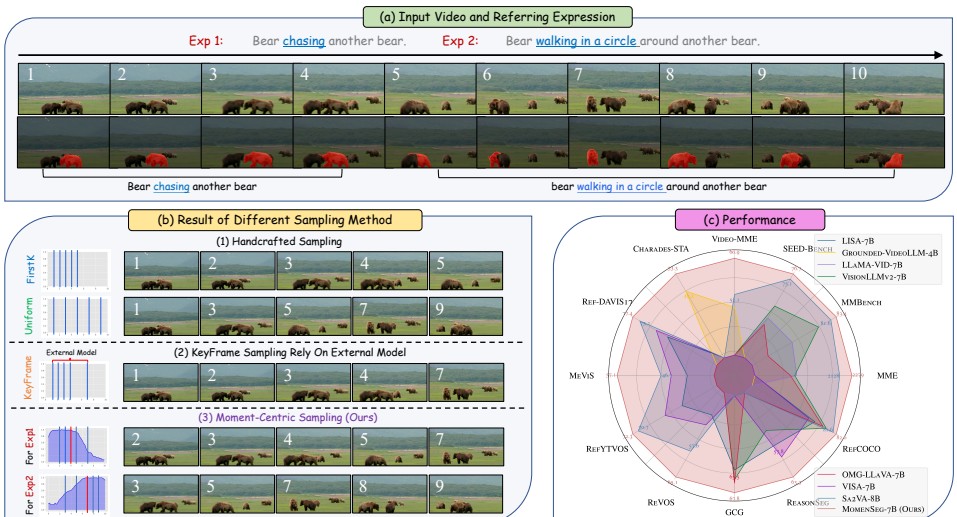

Figure 1: **Illustration of Moment-Centric Sampling.** (a) Two example expressions with 10 selected video frames and GT masks. (b) Comparison of sampling strategies: handcrafted, external keyframe–based, and our proposed *Moment-Centric Sampling* (MCS). MCS performs dense sampling at critical moments and sparse sampling elsewhere, without relying on external keyframe models. (c) MomentSeg achieves superior performance across RefVOS, TSG, and QA tasks compared with other LMM-based methods.

## Abstract

Referring Video Object Segmentation (RefVOS) seeks to segment target objects in videos guided by natural language descriptions, demanding both temporal reasoning and fine-grained visual comprehension. Existing sampling strategies for LLM-based approaches typically rely on either handcrafted heuristics or external keyframe models. The former often overlooks essential temporal cues, while the latter increases system complexity. To address this, we propose a unified framework that jointly optimizes Temporal Sentence Grounding (TSG) and RefVOS, naturally incorporating key moment grounding capability. During training, we introduce a novel TSG paradigm that employs a dedicated [FIND] token for key moment identification through temporal token similarity matching, thereby avoiding the need for external timestamp encodings. For inference, we design a Moment-Centric Sampling (MCS) strategy that densely samples informative moments while sparsely sampling non-essential frames, preserving both motion details and global context. To further enhance tracking stability, we develop Bidirectional Anchor-updated Propagation (BAP), which leverages the most relevant moment as start point for high-quality mask initialization and dynamically updates at sampled points to mitigate accumulated errors.

## 1 Introduction

RefVOS has attracted considerable attention in recent years. The task involves localizing and segmenting target objects at the pixel level in videos based on natural language expressions (Gavrilyuk

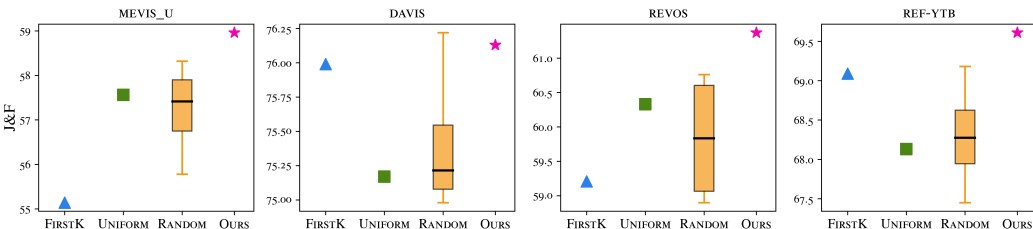

Figure 2: **Impact of Sampling Strategies.** We evaluate various sampling strategies for the RefVOS task on four datasets. Five independent runs of random sampling show high variance, highlighting the need for a robust sampling mechanism. Our proposed MCS consistently outperforms alternative methods across all datasets.

et al., 2018; Khoreva et al., 2019; Seo et al., 2020). It requires models to interpret temporal action semantics described in text and to maintain consistent tracking of the referred object across frames. Recent Transformer-based methods (Botach et al., 2022; Luo et al., 2023; He & Ding, 2024; Pan et al., 2025) mainly focus on improving temporal reasoning and inter-frame consistency. In contrast, large multimodal models (LMMs) (Bai et al., 2025; Chen et al., 2024f; Li et al., 2024a), pre-trained on massive video-text corpora, demonstrate stronger potential for advancing pixel-level video understanding tasks (Munasinghe et al., 2024; Lin et al., 2025; Yuan et al., 2025).

A central challenge in RefVOS lies in effectively sampling keyframes that are relevant to the linguistic expression. Given the variable and often large number of frames, feeding all frames into the model is both computationally prohibitive and redundant. Consequently, identifying and selecting semantically meaningful keyframes is worth exploring. Existing approaches employ diverse strategies for keyframe selection, as summarized in Table 1. For instance, Sa2VA (Yuan et al., 2025) simply selects the first K frames, disregarding content-specific cues such as object presence or action timing. VideoLISA (Bai et al., 2024) compresses temporal information and samples frames uniformly to retain coarse timing information but still overlooks referentially critical frames. In contrast, methods such as VISA (Yan et al., 2024), ViLLA (Zheng et al., 2024), VRS-HQ (Gong et al., 2025), and GLUS (Lin et al., 2025) depend on separately trained models to pinpoint the most relevant timestamps, then either use tracking networks (Cheng & Schwing, 2022) to generate mask trajectories or select key sampling frames to guide large-model segmentation. While these methods enable the identification of keyframes, they are dependent on these additional models.

Building on prior explorations, we first pose a fundamental question: ***Is keyframe selection necessary?*** From an ***intuitive*** perspective, failing to capture critical moments in a video risks missing the target object's appearance or the described action, which directly hinders accurate segmentation. Conversely, precisely identifying and leveraging text-relevant key moments mitigates the misleading effects of irrelevant frames on LMMs while reducing computational redundancy. From a ***quantitative*** perspective, as shown in Fig. 2, we evaluate several sampling strategies across multiple RefVOS datasets. Two consistent patterns emerge: (**1**) random sampling exhibits high variance, highlighting the strong influence of sampling strategy; and (**2**) for scenarios involving motion-driven (e.g., MeVIS (Ding et al., 2023)) and complex semantic understanding (e.g., ReVOS (Yan et al., 2024)), uniform sampling often surpasses the firstK approach. In contrast, firstK performs better on datasets (Khoreva et al., 2019; Seo et al., 2020) where targets appear early in the video and descriptions rely less on temporal cues. Likewise, MeVIS, which includes numerous action-centric descriptions and scenes with multiple visually similar objects, poses a particular challenge: if sampled frames miss the described action while static cues match several candidates, the model struggles to disambiguate the target. This analysis underscores that *keyframe selection is critical for RefVOS*.

Next, we consider the following question: ***How can we effectively identify keyframes, and how can these keyframes enhance RefVOS performance?*** Prior work offers two primary strategies: (**1**) Keyframe for tracking initialization. VISA (Yan et al., 2024) employs LLAMA-VID (Li et al., 2025b) to select the most critical frame as the tracking initialization timestamp, and applies XMem (Cheng & Schwing, 2022) for subsequent tracking. (**2**) Keyframes for LMM understanding. ViLLA (Zheng et al., 2024) leverages Grounded-VideoLLM (Wang et al., 2024a) to generate relevant temporal intervals and then samples frames inside and outside these intervals using handcrafted ratios, subsequently feeding these frames to LMM. VRS-HQ (Gong et al., 2025) further combines CLIP-based (Radford et al., 2021) vision-language similarity with SAM2 confidence scores to filter

Table 1: Comparison of *keyframe models*, *sampling*, and *information* utilized in existing RefVOS MLLMs. "cont." means continuous sampling. Unlike prior methods, our approach inherently supports keyframe selection while simultaneously leveraging both "Global" and "Essential" information during inference.

| Method | External Keyframe Model | Training | | Inference | | [SEG] |
|---|---|---|---|---|---|---|
| | | Sampling | Information | Sampling | Information | |
| VideoLISA (Bai et al., 2024) | / | Uniform | Global | Uniform | Global | 1 |
| VISA (Yan et al., 2024) | LLaMA-VID (Li et al., 2025b) | Random | Random | Uniform + Cont. | Global + Local | 1 |
| ViLLa (Zheng et al., 2024) | G-VideoLLM (Wang et al., 2024a) | Uniform | Global | Cont. | Local | $N$ |
| GLUS (Lin et al., 2025) | Chat-UniVi (Jin et al., 2024) | Uniform + Cont. | Global + Local | Uniform + Cont. | Global + Local | $N$ |
| Sa2VA (Yuan et al., 2025) | / | Random | Random | FirstK | Local | 1 |
| VRS-HQ (Gong et al., 2025) | CLIP (Radford et al., 2021) | Uniform | Global | Selection | Essential | $N$ |
| **MomentSeg (Ours)** | Inherently | Uniform + Cont. | Global + Local | Selection | Global + Essential | 1 |

keyframes before passing them to the LMM. While effective, all these methods depend on external models for keyframe identification, inevitably increasing overall system complexity.

In this paper, we propose a unified framework for the synchronized optimization of TSG and RefVOS. ***During training***, as shown in Fig. 3, we introduce a special [FIND] token for the TSG task, similar to the [SEG] token in RefVOS. This token computes similarity with sampled video frame tokens, offering two key advantages: **First**, for RefVOS, it eliminates the need for external models for keyframe selection, as the model natively identifies key moments. **Second**, it obviates the need for explicit timestamp encoding (Chen et al., 2024d) by leveraging a frame-level matching mechanism between [FIND] and temporal tokens. ***During inference***, as depicted in Fig. 4, we propose a Moment-Centric Sampling (MCS). MCS leverages the similarity distribution from the [FIND] token to densely sample video moments highly relevant to the description, while sparsely sampling less critical frames. This approach effectively preserves crucial motion cues while maintaining a global temporal context. For the RefVOS task, consistent with Sa2VA (Yuan et al., 2025), a single [SEG] token is used to represent the referential object throughout the video. We further introduce a novel Bidirectional Anchor-updated Propagation (BAP) strategy, which is composed of two key components: **(1)** *Bidirectional Propagation*: The central keyframes identified by MCS serve as starting points for propagation in both forward and backward temporal directions. **(2)** *Anchor-updated*: At these sampled time points, the target mask is updated and the prior memory is adaptively refreshed, which mitigates cumulative tracking errors.

In summary, our key contributions are as follows: **(1)** We propose **MomentSeg**, a unified framework that integrates temporal sentence grounding (TSG) and referring video object segmentation (RefVOS). For TSG, MomentSeg obviates explicit timestamp encoding by leveraging the similarity between the [FIND] token and temporal tokens. For RefVOS, our model inherently identifies text-relevant keyframes, thereby removing the dependency on external keyframe-selection models. **(2)** We introduce **Moment-Centric Sampling (MCS)**, a similarity-driven sampling strategy that densely samples frames around relevant moments and sparsely samples non-essential frames. MCS preserves salient motion cues while retaining global temporal context. **(3)** We present **Bidirectional Anchor-updated Propagation (BAP)**, which initiates mask propagation from query-relevant keyframes and adaptively updates the target mask at sampled temporal points. BAP enhances tracking robustness and mitigates cumulative propagation errors. **(4)** MomentSeg achieves new SOTA performance, delivering a 5% improvement on the MeVIS and a 6% gain on the ReVOS.

## 2 RELATED WORK

**Referring Video Object Segmentation.** RefVOS aims to segment targets in a video conditioned on a given description. Existing approaches typically employ object queries to fuse expressions with visual features for referent identification (Wu et al., 2022; 2023a; Luo et al., 2023; Tang et al., 2023; Yuan et al., 2024), while a line of work emphasizes motion aggregation to capture dynamic cues (Ding et al., 2023; He & Ding, 2024; Fang et al., 2025b). Concurrently, LMMs (Liu et al., 2024b) have been used to reason over compositional and complex expressions (Zhu et al., 2023; Yan et al., 2024; Bai et al., 2024; Munasinghe et al., 2024; Lin et al., 2025; Deng et al., 2025). To address these gaps, we propose **MomentSeg**, which performs keyframe selection natively, removing the need for external modules. We further introduce Bi-directional Anchor-updated Propagation (BAP), which propagates masks forward and backward from key moments and adaptively updates memory at sampled nodes, enhancing segmentation robustness and reducing cumulative errors.

**LLM-based RefVOS Sampling Strategies.** Existing LLM-based RefVOS methods adopt diverse frame-sampling and keyframe-selection schemes (Table 1). VideoLISA (Bai et al., 2024) and Sa2VA (Yuan et al., 2025) adopt handcrafted sampling strategies, using uniform or firstK sampling to generate a global [SEG] token. In contrast, VISA (Yan et al., 2024), ViLLA (Zheng et al., 2024), and GLUS (Lin et al., 2025) rely on external models (e.g., LLaMA-VID (Li et al., 2025b), Grounded-VideoLLM (Wang et al., 2024a), Chat-UniVi (Jin et al., 2024)) for keyframe selection, while VRS-HQ (Gong et al., 2025) selects keyframes via CLIP-based vision–language similarity (Radford et al., 2021). Such dependencies increase pipeline complexity and limit adaptability. By comparison, **MomentSeg** performs keyframe selection natively and introduces Moment-Centric Sampling (MCS), which applies dense sampling around text-relevant moments and sparse sampling elsewhere, preserving salient motion cues while maintaining a compact global temporal context.

**Timestamps Encoding in Temporal Sentence Grounding.** TSG (Gao et al., 2017; Caba Heilbron et al., 2015; Zala et al., 2023; Wang et al., 2024a; Li et al., 2025a; Liu et al., 2025a) aims to localize a video segment corresponding to a language query. A key challenge for LMM-based methods lies in developing an effective timestamp encoding strategy. Approaches such as TimeChat (Ren et al., 2024a), VTimeLLM (Huang et al., 2024a), and LITA (Huang et al., 2024b) leverage instruction tuning for plain text temporal grounding. Momentor (Qian et al., 2024a) injects explicit temporal position encodings into frame-level features to improve temporal localization, while VTG-LLM (Guo et al., 2024) incorporates absolute-time tokens. Vid2Seq (Yang et al., 2023) and Grounded-VideoLLM (Wang et al., 2024a) adopt discrete or relative temporal tokens to circumvent direct timestamp encoding. NumberIT (Wu et al., 2025) injects sequential frame indices into images for grounding. In contrast, **MomentSeg** avoids timestamp encoding by employing a specialized [FIND] token that performs frame-level temporal matching.

## 3 METHODS

### 3.1 PRELIMINARIES

**Referring Video Object Segmentation:** Given an input video with $T$ frames $I_{1:T} \in \mathbb{R}^{T \times H \times W \times 3}$ and a referring language expression $R$, RefVOS aims to train a model $\phi_\theta$ that predicts binary segmentation masks $M_{1:T}$ for the referred object:

$$M_{1:T} = \phi_\theta(I_{1:T}, R). \tag{1}$$

**LMMs-based RefVOS Paradigm:** Recent LMM-based RefVOS approaches build upon image-level referring segmentation models such as LISA (Lai et al., 2024), which employ a dedicated [SEG] token to represent the target and a segmentation decoder Dec to generate masks:

$$[\text{SEG}] = \text{LMM}(I, R), \quad M = \text{Dec}(I, [\text{SEG}]). \tag{2}$$

This paradigm extends to videos by adopting a multi-image formulation, where the LMM processes multiple frames $I_{1:N}$ and segmentation tokens decode masks for each frame:

$$[\text{SEG}] = \text{LMM}(I_{1:N}, R), \quad M_{1:N} = \text{Dec}(I_{1:N}, [\text{SEG}]). \tag{3}$$

This equation exemplifies a "One Token Seg All" paradigm (Bai et al., 2024), in which a single [SEG] token represents the referential object throughout the video. Here, $N$ denotes the number of frames provided to the LMM, which is typically smaller than the full sequence length $T$.

### 3.2 MODEL ARCHITECTURE OF MOMENTSEG

#### 3.2.1 OVERALL PIPELINE

**Training Pipeline:** Fig. 3 presents the training paradigm of MomentSeg for both TSG and RefVOS. For RefVOS, we follow a scheme similar to Sa2VA (Yuan et al., 2025), while additionally incorporating low-resolution, uniformly sampled frames $I^l_{1:L}$ and high-resolution, densely sampled frames $I^h_{1:N}$ from temporal segments where the referential object is present. The sampling of these low-resolution frames is referred to as *Temporal Token Injecting* in this paper. Notably, the number of tokens introduced into the LMM remains minimal, at no more than 16 tokens per frame in our setting. The low-resolution frames $I^l_{1:L}$ supply global temporal context, whereas the high-resolution frames $I^h_{1:N}$ capture fine-grained spatial details. A SAM2 decoder is then applied to the

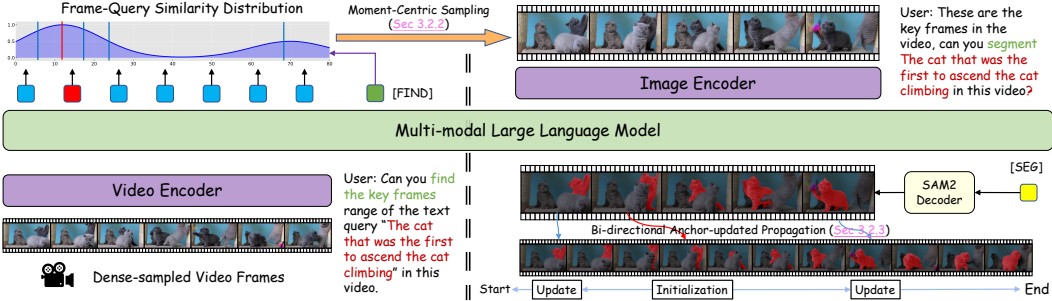

Figure 3: **Training framework of the proposed MomentSeg model**. In the TSG paradigm, we employ the Qwen2.5-VL (Bai et al., 2025) video encoder with low-resolution image inputs. We further introduce a [FIND] token trained under a contrastive learning scheme. In the RefVOS paradigm, both low-resolution uniformly sampled frames and high-resolution continuously sampled frames are used as inputs. Supervision is provided via the [SEG] token, with segmentation masks generated by the SAM2 decoder.

high-resolution frames and is supervised as follows:

$$[\text{SEG}]^h = \text{LMM}(I^l_{1:L}, I^h_{1:N}, R), \quad M_{1:N} = \text{Dec}(I^h_{1:N}, [\text{SEG}]^h). \quad (4)$$

For TSG, we introduce a special token [FIND] to identify relevant frames through similarity matching with temporal features. This design removes the need for external timestamp encoding, as temporal correspondence is established directly via frame-level matching. Specifically, we first extract the temporal tokens corresponding to the LMM output. These tokens are projected into a feature space using an MLP layer and subsequently average-pooled to obtain frame representations $T^v \in \mathbb{R}^{L_t \times C}$. The [FIND] tokens, $T^f \in \mathbb{R}^{N_f \times C}$ (where $N_f$ denotes the number of [FIND] tokens in a sample under the multi-round dialogue paradigm), are mapped through the same MLP layer. We then compute the similarity matrix $\ell \in \mathbb{R}^{N_f \times L_t}$ between frame tokens and [FIND] tokens to identify text-relevant frames. The matching loss is defined as:

$$\ell_{ij} = \frac{(T^f_i)^\top T^v_j}{\|T^f_i\| \, \|T^v_j\| \, \tau}, \mathcal{L}_{find} = \frac{1}{|\Omega|} \sum_{(i,j) \in \Omega} \Big[ -\lambda_p y_{ij} \log \sigma(\ell_{ij}) - (1 - y_{ij}) \log(1 - \sigma(\ell_{ij})) \Big], \quad (5)$$

where $\sigma$ denotes the sigmoid function, $T^v_i \in \mathbb{R}^C$ is the $i$-th pooled frame token, $T^f_j \in \mathbb{R}^C$ is the $j$-th [FIND] token, $y_{ij} \in \{0, 1\}$ is the match label, $\Omega$ is the set of valid pairs, and $\tau$ is the temperature factor, set to 0.07. The positive sample weight $\lambda_p$ is set to 2.0. Notably, since Qwen2.5-VL compresses the temporal dimension during video encoding (with $L_t$ being half of $L$), we apply bilinear interpolation to the similarity distribution to ensure alignment with the original input length.

Figure 4: **Inference framework of the proposed MomentSeg model.** First, we use dense-sampled video frames to find the frame-query similarity distribution related to the description, then apply a Moment-Centric Sampling (MCS) to select key frames from the sequence. These key frames are input to the model, along with the RefVOS instructions, to perform the RefVOS task. Finally, we enhance segmentation robustness through Bidirectional Anchor-updated Propagation (BAP).

**Inference Pipeline:** As shown in Fig. 4, inference begins with densely sampled low-resolution frames $I_{1:L}^l$ to compute the frame-query similarity distribution relative to the description. Based on this distribution, the Moment-Centric Sampling (MCS) strategy selects key frames $I_{1:N}^h$. These key frames, together with RefVOS instructions, are then input to the model for segmentation. Finally, Bidirectional Anchor-updated Propagation (BAP) is applied to enhance segmentation robustness.

### 3.2.2 MOMENT-CENTRIC SAMPLING (MCS)

MCS selectively samples text-relevant frames while preserving global temporal context. Specifically, we first quantify the semantic relevance of each frame by computing the similarity between temporal frame tokens $I_{1:L}^l$ and the [FIND] token $T^f$. Based on the resulting similarity distribution, intuitive strategies such as NearbyK and TopK sampling often lead to densely clustered selections. In contrast, MCS adaptively modulates sampling density according to frame-query similarity, concentrating coverage in high-relevance regions while maintaining a sparse representation elsewhere, as validated in Table 8. Concretely, Gaussian smoothing is applied to mitigate noise, yielding a refined similarity distribution $\mathcal{S} \in \mathbb{R}^T$. The moment center $c^*$ is then identified via a sliding window of size $w$ that maximizes the window-level cumulative similarity:

$$i^* = \arg\max_{i=0 \to T-w} \sum_{j=i}^{i+w-1} \mathcal{S}_j, \quad c^* = i^* + \lfloor \frac{w}{2} \rfloor, \tag{6}$$

where $i^*$ denotes the starting index of the optimal window. Alg. 1 details the process: *(1) Compute partitioned mass:* cumulative similarities $(w_L, w_R)$ are aggregated for regions flanking $c^*$. *(2) Allocate budget:* the remaining $K-1$ samples are distributed to the left $(k_L)$ and right $(k_R)$ partitions proportional to their weights. *(3) Perform stratified sampling:* InverseCDF is applied to each partition, and the resulting indices are combined with $c^*$ to form the final set. The InverseCDF procedure takes as input a sub-distribution $\mathcal{S}_{a:b}$ and a target sample size $k$. It normalizes the weights to form probabilities $p_i = \mathcal{S}_i / \sum_{j=a}^b \mathcal{S}_j$ and computes $F(i) = \sum_{j=a}^i p_j$. Then, the sampled indices are determined by querying uniformly spaced quantiles $\hat{u}_m \in (0,1)$:

---

**Algorithm 1** Moment-Centric Sampling

**input:** similarity scores $\mathcal{S} \in \mathbb{R}^T$, total samples $K \in \mathbb{R}^1$, moment center $c^* \in \mathbb{R}^1$
**output:** sampled indices $\mathcal{I} \subseteq \{0, \dots, T-1\}$
**(1)** *Compute partitioned similarity mass*
$\quad w_L \leftarrow \sum_{i=0}^{c^*-1} \mathcal{S}_i, \ \ w_R \leftarrow \sum_{i=c^*+1}^{T-1} \mathcal{S}_i.$
**(2)** *Allocate partition sampling budget*
$\quad k_L \leftarrow \left\lfloor (K-1)\frac{w_L}{w_L+w_R} \right\rceil,$
$\quad k_R \leftarrow K - 1 - k_L.$
**(3)** *Perform stratified sampling*
$\quad \mathcal{I}_L \leftarrow \text{InverseCDF}(\mathcal{S}_{0:c^*-1}, k_L),$
$\quad \mathcal{I}_R \leftarrow \text{InverseCDF}(\mathcal{S}_{c^*+1:T-1}, k_R).$
**return** $\text{sort}(\mathcal{I}_L \cup \{c^*\} \cup \mathcal{I}_R)$

---

$$i_m = \min\{ i \in [a,b] \mid F(i) \geq \hat{u}_m \}, \qquad \hat{u}_m = \frac{m-0.5}{k}, \qquad m = 1, \dots, k. \tag{7}$$

This ensures that moments with higher similarity receive denser sampling while still maintaining representation across the entire temporal span. The key advantage of MCS lies in its ability to enhance the capacity to capture both fine-grained actions and their broader temporal dependencies.

### 3.2.3 BIDIRECTIONAL ANCHOR-UPDATED PROPAGATION (BAP)

To improve segmentation robustness, we propose BAP, which leverages key moments as anchors for bidirectional propagation. The primary goal is to provide SAM2 with a strong initialization while ensuring stable tracking through mask updates during propagation. Its two key features are: (1) SAM2 propagation starts from critical temporal anchors and proceeds bidirectionally (*Initialization* in Fig. 4); and (2) mask updates occur at sampled key moments with selective memory clearing (*Update* in Fig. 4). This design enables dynamic error correction, enhancing long-term tracking. Specifically, the most relevant moment $c^*$ identified by MCS is selected as the starting point for bidirectional propagation. At each sampled key moment $\mathcal{I}$, a memory cleaning mechanism adaptively clears the cache based on the mask prediction score $S^p$ and the tracking score $S^t$, as determined by:

$$U_i = \begin{cases} 1, & \text{if } \prod_{j=1}^t S_j^t < \lambda \cdot S_i^p \\ 0, & \text{otherwise} \end{cases} \tag{8}$$

where $U_i$ is the binary indicator (1 for clearing memory, 0 for retention), $S_j^t$ is the tracking score at step $j$, $\prod_{j=1}^t S_j^t$ denotes the cumulative tracking confidence up to time $t$, $S_i^p$ is the prediction confidence at node $i$, and $\lambda$ is a sensitivity hyperparameter (set to 0.9). An update is triggered

Table 2: Comparison of MomentSeg with methods on temporal sentence grounding benchmarks.

| Method | Charades-STA | | | | ActivityNet-Grounding | | | |
|---|---|---|---|---|---|---|---|---|
| | R@0.3 | R@0.5 | R@0.7 | mIoU | R@0.3 | R@0.5 | R@0.7 | mIoU |
| VideoChat2-7B (Li et al., 2024b) | 38.0 | 14.3 | 3.8 | 24.6 | 40.8 | 27.8 | 9.3 | 27.9 |
| Momenter-7B (Qian et al., 2024b) | 42.6 | 26.6 | 11.6 | 28.5 | 42.9 | 23.0 | 12.4 | 29.3 |
| VTimeLLM-7B (Huang et al., 2024a) | 51.0 | 27.5 | 11.4 | 31.2 | 44.0 | 27.8 | 14.3 | 30.4 |
| TimeChat-7B (Ren et al., 2024a) | - | 32.2 | 13.4 | - | - | - | - | - |
| VTG-LLM-7B (Guo et al., 2025) | - | 33.8 | 15.7 | - | - | - | - | - |
| HawkEye-7B (Wang et al., 2024b) | 50.6 | 31.4 | 14.5 | 33.7 | 49.1 | 29.3 | 10.7 | 32.7 |
| Grounded-VideoLLM-4B (Wang et al., 2024a) | 54.2 | 36.4 | 19.7 | 36.8 | 46.2 | 30.3 | 19.0 | 36.1 |
| NumPro-LongVA-7B (Wu et al., 2025) | 63.8 | 42.0 | 20.6 | 41.4 | 55.6 | 37.5 | 20.6 | 38.8 |
| Qwen2.5-VL-7B (Bai et al., 2025) | - | - | - | 43.6 | - | - | - | - |
| **MomentSeg-3B (Ours)** | **76.1** | **58.2** | **25.8** | **50.2** | **67.5** | **44.7** | **23.2** | **45.4** |

Table 3: Comparison of MomentSeg with methods on referring video object segmentation benchmarks.

| Method | MeViS ($val^u$) | | | MeViS ($val$) | | | Ref-Youtube-VOS | | | Ref-DAVIS17 | | |
|---|---|---|---|---|---|---|---|---|---|---|---|---|
| | $\mathcal{J}\&\mathcal{F}$ | $\mathcal{J}$ | $\mathcal{F}$ | $\mathcal{J}\&\mathcal{F}$ | $\mathcal{J}$ | $\mathcal{F}$ | $\mathcal{J}\&\mathcal{F}$ | $\mathcal{J}$ | $\mathcal{F}$ | $\mathcal{J}\&\mathcal{F}$ | $\mathcal{J}$ | $\mathcal{F}$ |
| LISA-7B (Lai et al., 2024) | 43.2 | 39.9 | 46.5 | 37.2 | 35.1 | 39.4 | 53.9 | 53.4 | 54.3 | 64.8 | 62.2 | 67.3 |
| VideoLISA-3.8B (Bai et al., 2024) | 54.5 | 50.9 | 58.1 | 44.4 | 41.3 | 47.6 | 63.7 | 61.7 | 65.7 | 68.8 | 64.9 | 72.7 |
| VISA-7B (Yan et al., 2024) | - | - | - | 43.5 | 40.7 | 46.3 | 61.5 | 59.8 | 63.2 | 69.4 | 66.3 | 72.5 |
| ViLLa-6B (Zheng et al., 2024) | - | - | - | 49.4 | 46.5 | 52.3 | 67.5 | 64.6 | 70.4 | 74.3 | 70.6 | 78.0 |
| GLUS-7B (Lin et al., 2025) | 60.9 | - | - | 51.3 | 48.5 | 54.2 | 67.3 | 65.5 | 69.0 | - | - | - |
| InstructSeg-3B (Wei et al., 2025b) | - | - | - | - | - | - | 67.5 | 65.4 | 69.5 | 71.1 | 67.3 | 74.9 |
| Sa2VA-4B (Yuan et al., 2025) | - | - | - | 46.2 | - | - | 70.0 | - | - | 70.0 | - | - |
| Sa2VA-8B (Yuan et al., 2025) | - | - | - | 46.9 | - | - | 70.7 | - | - | 75.2 | - | - |
| Sa2VA-Qwen2.5-VL-3B (Yuan et al., 2025) | 57.5 | 53.6 | 61.3 | 50.0 | 46.9 | 53.1 | 71.0 | 68.8 | 73.1 | 74.4 | 70.1 | 78.7 |
| **MomentSeg-3B (Ours)** | 62.0 | 58.1 | 65.9 | 54.8 | 51.7 | 58.0 | 72.0 | 69.8 | 74.3 | 76.4 | 72.2 | 80.6 |
| **MomentSeg-7B (Ours)** | **62.6** | **58.7** | **66.5** | **57.1** | **53.9** | **60.2** | **72.3** | **70.1** | **74.5** | **77.4** | **73.2** | **81.7** |

when accumulated confidence falls below the threshold determined by the current prediction quality, ensuring that memory is refreshed only when beneficial for tracking accuracy.

# 4 EXPERIMENTS

## 4.1 IMPLEMENTATION DETAILS

We construct our baseline by integrating Qwen2.5-VL (Bai et al., 2025) with SAM2 (Ravi et al., 2024). Following prior work (Yuan et al., 2025), segmentation masks are generated by decoding the hidden state of the [SEG] token using the SAM2 decoder. The LLM is fine-tuned with LoRA (Hu et al., 2021), with a maximum sequence length of 8,192 tokens. Training is conducted on 8 NVIDIA H20 GPUs (96GB memory each) and takes approximately 24 hours for the MomentSeg-3B model. For evaluation, we adopt lmms-eval (Zhang et al., 2024a) for both image and video chat tasks.

## 4.2 MAIN RESULTS

**Temporal Sentence Grounding Task.** We evaluate our proposed method, MomentSeg, on the Charades-STA and ActivityNet datasets. As shown in Table 2, MomentSeg is compared with a range of advanced methods. The MomentSeg-3B model, despite its smaller scale, consistently outperforms most existing Video-LLMs. This strong performance validates the effectiveness of our proposed paradigm, which is underpinned by the novel mechanism where the [FIND] token interacts with temporal tokens via cosine similarity to achieve frame-level matching.

**Referring/Reasoning Video Segmentation Task.** For *referring* tasks (Table 3), MomentSeg-3B achieves $\mathcal{J}\&\mathcal{F}$ scores of 62.0 on MeViS ($val^u$) and 54.8 on MeViS ($val$), surpassing GLUS-7B by 1.1 and 3.5 points, respectively. On Ref-Youtube-VOS and Ref-DAVIS17, it attains scores of 72.0 and 76.4, improving over Sa2VA-4B by 2.0 and 6.4 points. Scaling to 7B further enhances performance across benchmarks. For *reasoning* tasks (Table 4), MomentSeg-3B obtains 62.6 on ReVOS, exceeding VRS-HQ-7B by 3.5

Table 5: Performance on Ref-SAV (Yuan et al., 2025) benchmark. FT means fine-tuning on Ref-SAV dataset.

| Method | Size | FT | $\mathcal{J}$ | $\mathcal{F}$ | $\mathcal{J}\&\mathcal{F}$ |
|---|---|---|---|---|---|
| UniRef++ (Wu et al., 2023b) | – | ✗ | 11.6 | 9.5 | 10.5 |
| UNINEXT (Yan et al., 2023) | – | ✗ | 8.8 | 6.4 | 7.6 |
| LMPM (Ding et al., 2023) | – | ✗ | 12.2 | 9.8 | 10.3 |
| VISA (Yan et al., 2024) | 7B | ✗ | 13.2 | 11.3 | 11.8 |
| Sa2VA (Yuan et al., 2025) | 8B | ✗ | 39.6 | 43.0 | 41.3 |
| UniPixel (Liu et al., 2025b) | 7B | ✗ | 68.5 | 69.6 | 69.0 |
| UniRef++ (Wu et al., 2023b) | – | ✓ | 15.8 | 13.4 | 14.6 |
| Sa2VA (Yuan et al., 2025) | 8B | ✓ | 48.3 | 51.7 | 50.0 |
| **MomentSeg-3B (Ours)** | 3B | ✗ | 79.3 | 80.6 | 79.9 |
| **MomentSeg-7B (Ours)** | 7B | ✗ | **80.1** | **81.4** | **80.8** |

Table 4: Comparison of MomentSeg with methods on reasoning video object segmentation benchmarks.

| Method | ReVOS | | | | | | | | | ReasonVOS | | |
| | Referring | | | Reasoning | | | Overall | | | | | |
| | $\mathcal{J}\&\mathcal{F}$ | $\mathcal{J}$ | $\mathcal{F}$ | $\mathcal{J}\&\mathcal{F}$ | $\mathcal{J}$ | $\mathcal{F}$ | $\mathcal{J}\&\mathcal{F}$ | $\mathcal{J}$ | $\mathcal{F}$ | $\mathcal{J}\&\mathcal{F}$ | $\mathcal{J}$ | $\mathcal{F}$ |
|---|---|---|---|---|---|---|---|---|---|---|---|---|
| TrackGPT-13B (Zhu et al., 2023) | 49.5 | 48.3 | 50.6 | 40.5 | 38.1 | 42.9 | 45.0 | 43.2 | 46.8 | - | - | - |
| VISA-13B (Yan et al., 2024) | 57.4 | 55.6 | 59.1 | 44.3 | 42.0 | 46.7 | 50.9 | 48.8 | 52.9 | - | - | - |
| VideoLISA-3.8B (Bai et al., 2024) | - | - | - | - | - | - | - | - | - | 47.5 | 45.1 | 49.9 |
| HyperSeg-3B (Wei et al., 2025a) | 58.5 | 56.0 | 60.9 | 53.0 | 50.2 | 55.8 | 55.7 | 53.1 | 58.4 | - | - | - |
| ViLLa-6B (Zheng et al., 2024) | - | - | - | - | - | - | 57.0 | 54.9 | 59.1 | 55.4 | - | - |
| GLUS-7B (Lin et al., 2025) | 58.3 | 56.0 | 60.7 | 51.4 | 48.8 | 53.9 | 54.9 | 52.4 | 57.3 | 49.9 | 47.5 | 52.4 |
| Sa2VA-4B (Yuan et al., 2025) | - | - | - | - | - | - | 53.2 | - | - | - | - | - |
| VRS-HQ-7B Gong et al. (2025) | 62.1 | 59.8 | 64.5 | 56.1 | 53.5 | 58.7 | 59.1 | 56.6 | 61.6 | - | - | - |
| Sa2VA-Qwen2.5-VL-3B (Yuan et al., 2025) | 61.5 | 59.0 | 64.1 | 55.9 | 53.0 | 58.8 | 58.7 | 56.0 | 61.4 | 49.6 | 46.9 | 52.4 |
| **MomentSeg-3B (Ours)** | 65.4 | 63.0 | 67.9 | 59.9 | 57.1 | 62.6 | 62.6 | 60.0 | 65.2 | 61.7 | 58.2 | 65.3 |
| **MomentSeg-7B (Ours)** | **67.4** | **64.9** | **69.8** | **62.8** | **59.8** | **65.7** | **65.1** | **62.3** | **67.8** | **62.7** | **59.2** | **66.1** |

Table 6: Comparison of model results across image segmentation tasks. Both RefCOCO/+/g and ReasonSeg use the cIoU metric, while GCG employs the mIoU metric.

| Method | RefCOCO | | | RefCOCO+ | | | RefCOCOg | | ReasonSeg | | GCG | |
| | val | testA | testB | val | testA | testB | val(U) | test(U) | val | test | val | test |
|---|---|---|---|---|---|---|---|---|---|---|---|---|
| PixelLM-7B (Ren et al., 2024b) | 73.0 | 76.5 | 68.2 | 66.3 | 71.7 | 58.3 | 69.3 | 70.5 | - | - | 62.0 | 61.7 |
| LISA-7B (Lai et al., 2024) | 74.9 | 79.1 | 72.3 | 65.1 | 70.8 | 58.1 | 67.9 | 70.6 | 61.3 | 62.9 | 62.0 | 61.7 |
| GLaMM-7B (Rasheed et al., 2024) | 79.5 | 83.2 | 76.9 | 72.6 | 78.7 | 64.6 | 74.2 | 74.9 | | | 65.8 | 64.6 |
| VisionLLMv2-7B (Wu et al., 2024) | 79.2 | 82.3 | 77.0 | 68.9 | 75.8 | 61.8 | 73.3 | 74.8 | 56.9 | 48.3 | 64.6 | 62.8 |
| OMG-LLaVA-7B (Zhang et al., 2024b) | 75.6 | 77.7 | 71.2 | 65.6 | 69.7 | 58.9 | 70.7 | 70.2 | - | - | 65.5 | 64.7 |
| LaSagnA-7B (Wei et al., 2024) | 76.8 | 78.7 | 73.8 | 66.4 | 70.6 | 60.1 | 70.6 | 71.9 | 48.8 | 47.2 | - | - |
| VISA-7B (Yan et al., 2024) | 72.4 | 75.5 | 68.1 | 59.8 | 64.8 | 53.1 | 65.5 | 66.4 | 52.7 | 57.8 | - | - |
| VRS-HQ-7B (Gong et al., 2025) | 73.5 | 77.5 | 69.5 | 61.7 | 67.6 | 54.3 | 66.7 | 67.5 | 51.8 | 52.9 | - | - |
| Sa2VA-4B (Yuan et al., 2025) | 78.9 | - | - | 71.7 | - | - | 74.1 | - | - | - | - | - |
| **MomentSeg-3B (Ours)** | 82.1 | 83.7 | 79.2 | 76.9 | 81.1 | 71.8 | 78.8 | 79.2 | 62.0 | 64.3 | 67.0 | 65.9 |
| **MomentSeg-7B (Ours)** | **82.6** | **85.1** | **80.2** | **78.2** | **81.9** | **72.3** | **80.1** | **80.1** | **63.3** | **65.5** | **67.8** | **67.9** |

points. On ReasonVOS, it achieves 61.7, outperforming ViLLa-6B by 6.3 points, with consistent improvements at larger scales. Furthermore, as shown in Table 5, on the Ref-SAV dataset, MomentSeg-7B achieves a $\mathcal{J}\&\mathcal{F}$ score of 80.0 without fine-tuning, exceeding the previous best UniPixel-7B method by 11.8 points. The evaluation set is from UniPixel Liu et al. (2025b).

**Referring/Reasoning Image Segmentation Task.** As shown in Table 6, MomentSeg-3B achieves 82.1, 76.9, and 78.8 on the RefCOCO/+/g validation sets, outperforming Sa2VA-4B by 3.2, 5.2, and 4.7 points, respectively. For reasoning tasks, MomentSeg-7B attains 65.5 on the ReasonSeg test set, surpassing the previous method, LISA-7B, by 2.6 points. Furthermore, on the GCG benchmark, MomentSeg-7B achieves 67.8/67.9 mIoU (val/test), exceeding GLaMM-7B by 2.0/3.3 points.

### 4.3 ABLATIONS STUDIES

**Key Components Design.** To assess the effectiveness of individual components, we conduct ablation studies on six RefVOS datasets. As shown in Table 7, starting from the baseline, introducing Temporal Token Injecting (TTI) consistently improves performance, with smaller gains on Ref-Youtube-VOS and Ref-DaVIS17 due to shorter videos and static expression. Incorporating MCS provides further improvements, particularly on motion-intensive datasets such as MeViS, where key frame sampling enhances action understanding. Finally, adding BAP strengthens tracking and error correction, with the largest benefits observed on reasoning-oriented benchmarks involving long video sequences, underscoring the importance of robust temporal tracking mechanisms.

**Effectiveness of Moment-Centric Sampling.** Sampling strategies impact performance in a dataset-specific manner. MeViS requires action understanding in long videos where targets appear only in specific segments, Ref-DAVIS17 contains targets present throughout most of the video, and Reason-VOS emphasizes relational reasoning with targets often emerging mid-video. As shown in Table 8, *FirstK* underperforms relative to *Uniform* on MeViS and ReasonVOS, as early frames frequently omit target segments. Keyframe-based strategies mitigate this by prioritizing high-similarity moments: *NearbyK* samples adjacent frames, while *TopK* selects frames with globally highest similarity. Both strategies rely on the similarity distribution from [FIND], yet they tend to concentrate samples in highly correlated contiguous regions, leading to redundancy in local frames and loss of global context. This limitation is particularly pronounced for long videos with complex textual descriptions. Motivated by this, our proposed **MCS** combines dense sampling near key moments with

Table 7: Effectiveness of the key components. We report the $\mathcal{J}\&\mathcal{F}$ metric for all six datasets.

| ID | TTI | MCS | BAP | MeViS ($val^u$) | MeViS ($val$) | Ref-DAVIS17 | Ref-Youtube-VOS | ReasonVOS | ReVOS (Overall) |
|---|---|---|---|---|---|---|---|---|---|
| 1 | | | | 57.0 | 51.3 | 75.5 | 70.1 | 54.1 | 58.3 |
| 2 | ✓ | | | 58.7(↑1.7) | 52.3(↑1.0) | 76.2(↑0.7) | 70.3(↑0.2) | 58.1(↑4.0) | 59.2(↑0.9) |
| 3 | ✓ | ✓ | | 61.3(↑2.6) | 53.9(↑1.6) | 76.8(↑0.6) | 70.8(↑0.5) | 60.8(↑2.7) | 60.4(↑1.2) |
| 4 | ✓ | ✓ | ✓ | 62.0(↑0.7) | 54.1(↑0.2) | 76.4(↓0.4) | 72.0(↑1.2) | 61.7(↑0.9) | 62.6(↑2.2) |

Table 8: Impact of moment-centric sampling.

| Method | MeViS ($val^u$) | Ref-DAVIS17 | ReasonVOS |
|---|---|---|---|
| FirstK (Baseline) | 58.3 | 76.3 | 55.9 |
| Uniform | 59.6(↑1.3) | 76.2(↓0.1) | 57.1(↑1.2) |
| KeyFrame (TopK) | 60.1 | 76.9 | 60.2 |
| KeyFrame (NearbyK) | 60.9(↑0.8) | 77.1(↑0.2) | 59.9(↓0.3) |
| MCS | 61.3(↑3.0) | 77.3(↑1.0) | 60.8(↑4.9) |

Table 9: Effectiveness of bidirectional anchor-updated propagation. $\mathcal{J}\&\mathcal{F}$ metrics are reported.

| Method | MeViS ($val^u$) | Ref-DAVIS17 | ReasonVOS |
|---|---|---|---|
| Forward Propagation (Baseline) | 58.9 | 76.2 | 57.4 |
| +Moment-anchored Updating | 61.1(↑2.2) | 76.9(↑0.7) | 60.8(↑3.4) |
| +Adaptive Memory Cleaning | 61.3(↑0.2) | 77.0(↑0.1) | 61.0(↑0.2) |
| +Bidirectional Propagation | 62.0(↑0.7) | 76.8(↓0.2) | 61.7(↑0.7) |

Table 10: Effectiveness of temporal token injecting.

| Method | MeViS ($val^u$) | Ref-DAVIS17 | ReasonVOS |
|---|---|---|---|
| *Training Process* | | | |
| w/o Temporal Tokens | 59.1 | 73.3 | 53.1 |
| w. Temporal Tokens | 60.1 (↑1.0) | 74.2(↑0.9) | 55.1(↑2.0) |
| *Inference Process* | | | |
| w/o Temporal Tokens | 60.1 | 74.2 | 55.1 |
| w. Temporal Tokens | 60.3 (↑0.2) | 73.2(↓1.0) | 56.4(↑1.3) |

Table 11: Effectiveness of joint training across tasks.

| Training Data | | | RefVOS | | TSG |
|---|---|---|---|---|---|
| RefVOS | TSG | VideoChat | MeViS | ReVOS | Charades-STA |
| ✓ | | | 50.7 | 58.9 | - |
| | ✓ | | - | - | 47.7 |
| ✓ | ✓ | | 51.4(↑0.7) | 59.8(↑0.9) | 48.9(↑1.2) |
| ✓ | ✓ | ✓ | 52.7(↑1.3) | 60.3(↑0.5) | 49.3(↑0.4) |

sparse sampling elsewhere, effectively capturing both crucial local cues and global context. This approach achieves the best overall performance across benchmarks, yielding improvements of 3.0 and 4.9 percentage points over *FirstK* on MeViS and ReasonVOS, respectively. Importantly, MCS introduces negligible computational overhead, contributing less than 0.1% to the total latency.

**Effectiveness of Bidirectional Anchor-updated Propagation.** Mask propagation often struggles with poor initialization when the target appears late in the video. BAP overcomes this by initializing at the most probable target moment, propagating bidirectionally, and applying adaptive memory updates to limit error accumulation. Table 9 shows ablations of its components. *Moment-anchored Updating* notably improves performance, particularly on long video scenarios, by correcting accumulated tracking errors, while *Adaptive Memory Cleaning* provides smaller but consistent improvements. Overall, *Bidirectional Propagation* strategy improves robustness by initializing from the most probable target moment without adding inference cost.

**Effectiveness of Temporal Tokens for TSG and RefVOS.** We use temporal tokens in training for both tasks, with RefVOS additionally sampling frames for segmentation. We examine: **(1)** whether TTI in RefVOS establishes temporal action cues; **(2)** whether tokens can be omitted at inference to reduce overhead; and **(3)** whether jointly optimizing temporal tokens for RefVOS and TSG leads to mutual benefits. For (1) and (2), Table 10 shows consistent gains across datasets, including +2 points on ReasonVOS. Notably, inference without tokens achieves similar performance, so we omit them to avoid extra cost. More detailed analyses are in Appendix E.1. For (3), Table 11 demonstrates that joint training improves both tasks, yielding a 1.2-point gain on TSG. Jointly optimizing temporal tokens provides mutual benefits, and incorporating the video-chat data further boosts performance.

Table 12: Comparison of inference efficiency and performance. The "Latency" denotes the end-to-end inference time per video, measured on a single NVIDIA A100 GPU (40 GB).

| Method | Ref-Youtube-VOS | | MeVIS ($val$) | | ReVOS | |
|---|---|---|---|---|---|---|
| | $\mathcal{J}\&\mathcal{F}$ | Latency (s) | $\mathcal{J}\&\mathcal{F}$ | Latency (s) | $\mathcal{J}\&\mathcal{F}$ | Latency (s) |
| Sa2VA-8B | 70.7 | 6.8 | 46.9 | 7.5 | 57.6 | 6.3 |
| **MomentSeg-7B (Ours)** | 72.3(↑1.6) | 7.0(↓3%) | 57.1(↑10.2) | 7.7(↓3%) | 65.1(↑7.5) | 6.4(↓2%) |
| ↪ Preprocess | - | 1.8 (26%) | - | 3.3 (43%) | - | 2.4 (38%) |
| ↪ MLLM | - | 1.3 (19%) | - | 1.4 (18%) | - | 1.3 (20%) |
| ↪ TSG | - | 0.5 (7%) | - | 0.6 (8%) | - | 0.5 (8%) |
| ↪ RefVOS | - | 0.8 (12%) | - | 0.8 (10%) | - | 0.8 (12%) |
| ↪ SAM2 | - | 3.9 (55%) | - | 3.0 (39%) | - | 2.7 (42%) |

**Analysis of Inference Latency.** As shown in Table 12, MomentSeg achieves notable performance gains on the RefVOS task (+10% on MeVIS and +7% on ReVOS) with only a slight increase in inference latency (approximately 3%). To analyze latency sources, we divide the process into three stages: *Preprocessing*, *MLLM*, and *SAM2*. The runtime of Preprocessing and SAM2 varies across datasets due to differences in video length and resolution, while MLLM remains stable, accounting for about 20% of total latency. The MLLM overhead in MomentSeg is slightly higher because

the TSG stage introduces an additional 7% computation cost, which is still minor compared with SAM2 propagation. This overhead can be further reduced by decreasing the number or resolution of temporal frames in the TSG stage.

**Analysis of Robustness.** We evaluate the robustness of BAP on the MeViS ($val^u$) dataset using four metrics: *Initial Shot Ratio*, *Initial mIoU*, *Anchor mIoU* with/without updating, and *Correction Ratio*. As shown in Table 13, MomentSeg significantly improves both the Initial Shot Ratio and Initial mIoU by leveraging MCS to select more reliable starting frames. Moreover, anchor updating increases the anchor-frame mIoU by 2.6 points compared with the non-updating setting, highlighting its effective correction ability. Overall, BAP corrects about 7% of low-IoU cases, demonstrating its robustness and practical benefit.

Table 13: Robustness evaluation of BAP on the MeViS ($val^u$) dataset. Initial Shot Ratio denotes the percentage of initial frames containing the referred target. Initial mIoU measures the segmentation quality on the initial frame. Anchor mIoU compares performance without and with anchor updating. Correction Ratio indicates the proportion of frames with IoU < 0.3 that are corrected to IoU > 0.7 via updating.

| Method | Initial Shot Ratio (%) | Initial mIoU (%) | Anchor mIoU (w/o → w/ Updating) (%) | Correction Ratio (%) |
|---|---|---|---|---|
| Sa2VA-8B | 88.7 | 54.0 | – | – |
| **MomentSeg-7B (Ours)** | 96.6 (↑7.9) | 63.3 (↑9.3) | 61.0 → 63.6 (↑2.6) | 7.1 |

## 5 QUALITATIVE RESULTS

As illustrated in Fig. 5, we show qualitative results of MomentSeg on TSG and RefVOS tasks. For the referring task (Fig. 5a), the model accurately localizes the described action and, through dense moment sampling with MCS, delivers precise segmentation. For the reasoning task (Fig. 5b), MomentSeg identifies the temporal segment aligned with complex relational descriptions, highlighting its strong temporal reasoning. Additional visualizations are provided in Appendix F.

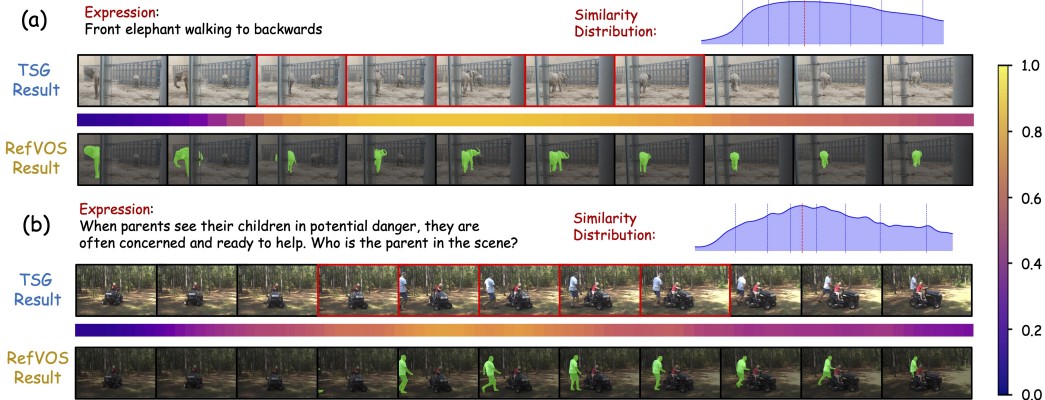

Figure 5: Qualitative results of MomentSeg on TSG and RefVOS tasks. The figure displays the input expression, frame-query similarity distribution, sampled frames, and the predicted segmentation masks for TSG and RefVOS. (a) illustrates an example for the referring task, while (b) presents an example for the reasoning task.

## 6 CONCLUSION

In this paper, we introduce MomentSeg, a unified framework that optimizes both TSG and RefVOS tasks simultaneously. Our approach eliminates the reliance on external models for keyframe selection, addressing the inherent complexity of previous methods. By leveraging a `[FIND]` token during training, MomentSeg identifies text-relevant key moments directly, bypassing the need for explicit timestamp encodings and enhancing temporal understanding. For inference, we propose Moment-Centric Sampling (MCS), which intelligently selects relevant frames while preserving essential motion cues and global temporal context. Additionally, our Bidirectional Anchor-updated Propagation (BAP) further improves tracking robustness by mitigating cumulative errors and handling occlusions effectively. Experiments on several RefVOS and TSG benchmarks demonstrate that MomentSeg outperforms existing methods, achieving SOTA results.

## 7 ETHICS STATEMENT

This work focuses on algorithmic analysis and enhancements to improve the capabilities of LMMs for the referring video object segmentation (RefVOS) task. All datasets employed are publicly available and have undergone appropriate ethical review, ensuring lawful and responsible use.

## 8 REPRODUCIBILITY STATEMENT

To support reproducibility, we provide detailed implementation information and materials. Section 4.1 describes the construction of the baseline models, while Appendix B presents training configurations, datasets, and evaluation metrics. Appendix C.2 further details the templates and procedures used to generate training data instructions. The complete codebase and trained models will be released upon publication to facilitate community verification and replication.

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

APPENDIX

We provide an overview of the Appendix below. Each section includes a brief description of its contents.

## A  USE OF LARGE LANGUAGE MODELS (LLMS)

A Large Language Model was employed solely to improve the manuscript's language, including grammar, clarity, and readability. No technical content, experimental design, analysis, or conclusions were produced or altered by the LLM. All scientific ideas, methodologies, and results remain entirely the authors' original work.

## B  ADDITIONAL IMPLEMENTATION DETAILS

### B.1  DATASET AND EVALUATION METRICS

**Datasets:** We train our MomentSeg model on a comprehensive dataset encompassing five categories: temporal sentence grounding, image question answering (QA), video QA, image segmentation, and video segmentation. The detailed composition of the training set is provided in Table 14. Our data curation largely follows the methodology of Sa2VA (Yuan et al., 2025), with notable additions including ReasonSeg (0.2K) (Lai et al., 2024) and two temporal sentence grounding datasets: Charades-STA (5.3K) (Gao et al., 2017) and ActivityNet Captions (10K) (Caba Heilbron et al., 2015).

Given that the Qwen2.5-VL (Bai et al., 2025) has already been extensively pre-trained on image and video QA data, we deliberately restrict the inclusion of such data to 665K samples from LLaVA 1.5 (Liu et al., 2024a) and 100K from ChatUniVi (Jin et al., 2024), thereby mitigating the risk of catastrophic forgetting of its foundational QA capabilities. For image-level referring segmentation, we incorporate 56K referring expression data (Kazemzadeh et al., 2014; Yu et al., 2016) and 214K

grounding conversation generation data (Rasheed et al., 2024). For video-level referring segmentation, we utilize 5.8K existing referring VOS samples from Ref-YouTubeVOS (Seo et al., 2020), MeVIS (Ding et al., 2023), and ReVOS (Yan et al., 2024).

**Evaluation Metrics:** For RefVOS, unless specified otherwise, we use the following evaluation metrics: $\mathcal{J}$ (average Intersection over Union, IoU), $\mathcal{F}$ (boundary F-measure), and $\mathcal{J}\&\mathcal{F}$ (the average of $\mathcal{J}$ and $\mathcal{F}$). For TSG, we report the Intersection over Union (IoU) between the predicted and ground truth timestamps, including Recall at IoU thresholds of {0.3, 0.5, 0.7}, as well as the mean IoU.

Table 14: Composition of datasets from different tasks used during training.

| Type | Datasets |
|---|---|
| Temporal Sentence Grounding | Charades-STA (5.3K), ActivityNet Captions (10K) |
| Image QA | LLaVA 1.5 (665K) |
| Image Segmentation | RefCOCO (17K), RefCOCO+ (17K), RefCOCOg (22K), Grand-f (214K), ReasonSeg (0.2K) |
| Video QA | ChatUniVi (100K) |
| Video Segmentation | Ref-YTVOS (3.5K), MeVIS (0.6K), ReVOS (1.7K), Ref-SAV (37K) |

## B.2 Training Details

Table 15 summarizes the training configurations. Our model is trained on 8 H20 GPUs with a per-GPU batch size of 2, requiring approximately 24 hours for the 3B model. During training, we configure the maximum temporal frames to 50, TSG frames to 60, and video frames to 5. For inference, we increase the maximum TSG frames to 100, set the TSG threshold to 0.4, and use 8 video frames. The maximum pixel resolutions are configured as follows: video inputs at $16 \times 16 \times 28 \times 28$, temporal inputs at $4 \times 4 \times 28 \times 28$, TSG inputs at $8 \times 8 \times 28 \times 28$, and image inputs at $20 \times 20 \times 28 \times 28$. We employ the AdamW optimizer with $\beta_1 = 0.9$, $\beta_2 = 0.999$, and a weight decay of 0.05. The learning rate is set to $4 \times 10^{-5}$ with a warmup ratio of 0.05.

Table 15: Implementation details of training process.

| Config | Value |
|---|---|
| *max temporal frame num* | 50 |
| *max TSG frame num* | 60 |
| *video frame num* | 5 |
| *max video pixels* | 16*16*28*28 |
| *max temporal pixels* | 4*4*28*28 |
| *max TSG pixels* | 8*8*28*28 |
| *max image pixels* | 20*20*28*28 |
| *optimizer* | AdamW |
| *optimizer momentum* | $\beta_1, \beta_2 = 0.9, 0.999$ |
| *optimizer weight decay* | 0.05 |
| *learning rate* | 4e-5 |
| *LoRA rank* | 128 |
| *batch size* | 2 |
| *warmup ratio* | 0.05 |

## C Additional Methods

### C.1 Temporal Token Injecting

Beyond the visual tokens extracted from selected video frames, we introduce temporal tokens to explicitly encode temporal dynamics. These tokens are meticulously designed to capture the relative temporal ordering of frames within the video sequence, thereby substantially bolstering the model's capacity for temporal reasoning. Our methodology draws conceptual parallels with GLUS (Lin et al., 2025), which integrates a finite set of global tokens, and VideoLISA (Bai et al., 2024), which condenses video frames into a singular token representation. A key distinction lies in that our temporal tokens are exclusively injected during the training phase. Furthermore, we leverage the Qwen2.5-VL's video encoder mode to achieve significant computational efficiency. Specifically, these temporal tokens are generated from uniformly sampled video frames. Each sampled frame undergoes

compression to a highly compact input dimension, with a maximum allocation of 16 tokens per individual image. The max count of uniformly sampled frames is restricted to 50. This entire process can be formally described by Eq. 4.

## C.2 INSTRUCTION DETAILS

We list the prompts used for different tasks during training in Table 16. For the TSG task, the prompt consists of video, query, and answer components, where the video component is constructed using Qwen2.5-VL's `<|video_pad|>` placeholder. For the RefVOS task, the prompt consists of video, image, query, and answer components. Compared to TSG, RefVOS includes an additional image component, which is constructed using Qwen2.5-VL's `<|image_pad|>` placeholder. For the answer component, we use `[FIND]` and `[SEG]` to represent the answers for TSG and RefVOS tasks, respectively. We design multiple different prompts for each component and randomly sample them during training to improve the model's robustness.

# D  ADDITIONAL RESULTS

## D.1  RESULTS ON IMAGE/VIDEO QA TASKS.

Maintaining basic QA capabilities remains a significant challenge for LMMs designed for specific tasks. Table 17 presents the performance of our method on image/video-based question answering tasks (Fu et al., 2023; Chen et al., 2024c; Kembhavi et al., 2016). Compared to other MLLMs with segmentation capabilities, our approach achieves the best accuracy trade-off between image chat and referential segmentation datasets. Additionally, our method excels on the Video-MME (Fu et al., 2025) video question answering benchmark, demonstrating its strong capabilities in multimodal understanding. Furthermore, we replicate the results for Qwen2.5-VL-3B (Bai et al., 2025) using the lmms-eval framework (Zhang et al., 2024a), ensuring consistency and fairness in evaluation. After incorporating segmentation capabilities into MomentSeg based on Qwen2.5-VL, there is no significant degradation in the underlying QA performance. In fact, MomentSeg outperforms on certain evaluation sets, such as MMbench (Liu et al., 2024d), SEED-Bench (Li et al., 2023), and MMMU (Yue et al., 2023), showcasing improvements in specific benchmarks.

## D.2  COMPARISON OF IMAGE/VIDEO SEGMENTATION WITH NON-LLM-BASED METHODS

Recent advances in small-scale models have achieved remarkable progress in traditional RIS and RVOS tasks. Several approaches (Dai et al., 2024; 2025c) leverage powerful vision-language understanding models such as BEiT-3 (Wang et al., 2023), substantially enhancing referring comprehension capabilities in both image (Xiao et al., 2024; Dai et al., 2025a) and video (Rong et al., 2025; Zhang et al., 2024c) domains. To provide a comprehensive comparative analysis, we additionally evaluate non-LLM-based methods. As shown in Table 18, MomentSeg, as a general-purpose video and image segmentation model, demonstrates competitive performance across diverse tasks, particularly when compared to smaller-scale models. Notably, it excels on datasets requiring temporal action understanding, such as MeVIS, where it exhibits superior capabilities in modeling temporal dynamics.

# E  ADDITIONAL ABLATION STUDIES

## E.1  ANALYSIS OF TEMPORAL TOKENS INJECTING.

To examine the effect of temporal tokens on model performance, we conduct ablation studies by varying their number during both training and inference. As shown in Fig. 6, we report results on three RefVOS datasets: MeVIS, Ref-DAVIS17, and ReasonVOS. During training, increasing the number of temporal frames yields a steady performance gain on MeVIS, saturating around 50 frames. Training cost grows proportionally with the number of temporal frames; thus we set it to 50 in all experiments. At inference, as also illustrated in Fig. 7, adding temporal frames has minimal impact on accuracy. To improve efficiency, we therefore omit temporal frames during inference.

**Prompts for Temporal Sentence Grounding**

**Video Prompt:**

    (1) "`<video>`\n This is a low-resolution video."

    (2) "`<video>`\n You can view this low-resolution video for reference."

    (3) "`<video>`\n This is a low-resolution video for analysis."

    (4) "`<video>`\n Here is a low-resolution video you can refer to."

**Question Prompt:**

    (1) "Can you find the key frames range of the text query '`<query>`' in this video?"

    (2) "Could you identify the key frames range for the text query '`<query>`' in this low-resolution video?"

    (3) "Please locate the key frames range where the text query '`<query>`' appears in this video?"

    (4) "Can you determine the key frames range containing the text query '`<query>`' in this video?"

**Answer Prompt:**

    (1) "It is [FIND]."

    (2) "Sure, [FIND]."

    (3) "Sure, it is [FIND]."

    (4) "[FIND]."

**Prompts for Referring Video Object Segmentation**

**Video Prompt:**

    (1) "`<video>`\n This is a low-resolution video."

    (2) "`<video>`\n You can view this low-resolution video for reference."

    (3) "`<video>`\n This is a low-resolution video for analysis."

    (4) "`<video>`\n Here is a low-resolution video you can refer to."

**Question Prompt:**

    (1) "`<images>`\n Can you segment the `<query>` in this video?"

    (2) "`<images>`\n Please segment `<query>` in this video."

    (3) "`<images>`\n What is `<query>` in this video? Please respond with segmentation mask."

    (4) "`<images>`\n What is `<query>` in this video? Please output segmentation mask."

    (5) "`<images>`\n Could you provide a segmentation mask for the `<query>` in this video?"

    (6) "`<images>`\n Please identify and segment the `<query>` in this video."

    (7) "`<images>`\n Where is the `<query>` in this video? Please respond with a segmentation mask."

    (8) "`<images>`\n Can you highlight the `<query>` in this video with a segmentation mask?"

**Answer Prompt:**

    (1) "It is [SEG]."

    (2) "Sure, [SEG]."

    (3) "Sure, it is [SEG]."

    (4) "Sure, the segmentation result is [SEG]."

    (5) "[SEG]."

Table 16: Prompts used for different tasks.

Table 17: Performance on image/video benchmarks with MLLMs possessing segmentation capabilities. The results for Qwen2.5-VL-3B (Bai et al., 2025) are replicated using the same lmms-eval (Zhang et al., 2024a) framework.

| Method | Image QA | | | | | | Video QA |
|---|---|---|---|---|---|---|---|
| | MME | MMBench | SEED-Bench | AI2D | MMStar | MMMU | Video-MME |
| LISA-7B (Lai et al., 2024) | 1/1 | 0.4 | - | - | - | - | - |
| PixelLM-7B (Ren et al., 2024b) | 309/135 | 17.4 | - | - | - | - | - |
| LaSagnA-7B (Wei et al., 2024) | 0/0 | 0.0 | - | - | - | - | - |
| GLaMM-7B (Rasheed et al., 2024) | 14/9 | 36.8 | - | 28.2 | - | - | - |
| OMG-LLaVA-7B (Zhang et al., 2024b) | 1177/235 | 47.9 | 56.5 | 42.9 | - | - | - |
| Sa2VA-4B (Yuan et al., 2025) | 1553/540 | 76.8 | 72.6 | 79.9 | 53.7 | 46.2 | 50.4 |
| Sa2VA-8B (Yuan et al., 2025) | **1651**/578 | 82.4 | 75.5 | 82.1 | 60.3 | 44.7 | 52.1 |
| Qwen2.5-VL-3B (Bai et al., 2025) | 1511/622 | 77.6 | 74.7 | 78.4 | 55.7 | 46.4 | 57.7 |
| **MomentSeg-3B (Ours)** | 1491/529 | 79.2 | 74.8 | 78.3 | 56.0 | 47.6 | 55.4 |
| **MomentSeg-7B (Ours)** | 1595/**684** | **83.4** | **76.7** | **82.6** | **62.1** | **50.3** | **60.9** |

Table 18: Comparison of model results with non-LLMs-based methods

| Method | RefCOCO | RefCOCO+ | RefCOCO | MeVIS | Ref-Youtube-VOS | Ref-DAVIS17 |
|---|---|---|---|---|---|---|
| ReLA (Liu et al., 2023) | 73.8 | 66.0 | 65.0 | - | - | - |
| EEVG (Chen et al., 2024e) | 79.5 | 71.9 | 73.6 | - | - | - |
| C3VG (Dai et al., 2025d) | 81.4 | 77.1 | 76.3 | - | - | - |
| OneRef-L (Xiao et al., 2024) | 81.3 | 76.6 | 75.7 | - | - | - |
| DeRIS-B (Dai et al., 2025b) | 82.0 | 75.6 | 76.3 | - | - | - |
| SOC (Luo et al., 2023) | - | - | - | - | 67.3 | 65.8 |
| DsHmp (He & Ding, 2024) | - | - | - | 46.4 | 67.1 | 64.9 |
| DMVS (Fang et al., 2025a) | - | - | - | 48.6 | 64.3 | 65.2 |
| SAMWISE (Cuttano et al., 2025) | - | - | - | 48.3 | 67.2 | 68.5 |
| ReferDINO (Liang et al., 2025) | - | - | - | 49.3 | 69.3 | 68.9 |
| MPG-SAM2 (Rong et al., 2025) | - | - | - | 53.7 | **73.9** | 72.4 |
| **MomentSeg-3B (Ours)** | 82.1 | 76.9 | 78.8 | 54.8 | 72.0 | 76.4 |
| **MomentSeg-7B (Ours)** | **82.6** | **78.2** | **80.1** | **57.1** | 72.3 | **77.4** |

## E.2 [FIND] MATCHING PARADIGM V.S. PLAIN-TEXT GENERATION

To compare the effectiveness of the [FIND] token approach, we conducted an ablation experiment that simultaneously supports both matching paradigm and plain-text generation. In prompt construction, we additionally include output with format of (start_ratio, end_ratio) as a 0-1 range. after the [FIND] instruction. The answer construction further adds The range is ({},{}) based on the [FIND] token, enabling the model to perform TSG tasks through both methods simultaneously. The experimental results in Table 19 demonstrate that the [FIND] token approach significantly outperforms the plain-text generation method across all metrics on both Charades-STA and ActivityNet-Grounding datasets. On Charades-STA, the [FIND] token achieves improvements of 10.4, 11.2, and 6.5 points in R@0.3, R@0.5, and R@0.7, respectively, with a substantial 9.3-point gain in mIoU. Similarly, on ActivityNet-Grounding, the [FIND] token shows consistent improvements, particularly in R@0.3 and R@0.5 metrics. The superior performance of the [FIND] token can be attributed to its direct prediction mechanism. Unlike the plain-text generation approach, which requires the model to understand and generate explicit timestamp encoding information, the [FIND] token directly enables the model to predict text-relevant key positions through similarity-based localization. This approach significantly simplifies the learning difficulty by eliminating the need for complex temporal coordinate regression, allowing the model to focus on semantic alignment between textual queries and visual content rather than numerical timestamp generation.

Table 19: Effect of post-process threshold $\theta$ on TSG.

| Method | Charades-STA | | | | ActivityNet-Grounding | | | |
|---|---|---|---|---|---|---|---|---|
| | R@0.3 | R@0.5 | R@0.7 | mIoU | R@0.3 | R@0.5 | R@0.7 | mIoU |
| Plain-Text Generation | 64.9 | 45.4 | 25.8 | 41.3 | 65.1 | 48.6 | 27.9 | 46.0 |
| [FIND] Matching | 75.3 | 56.6 | 32.3 | 50.6 | 70.3 | 52.0 | 29.0 | 49.3 |

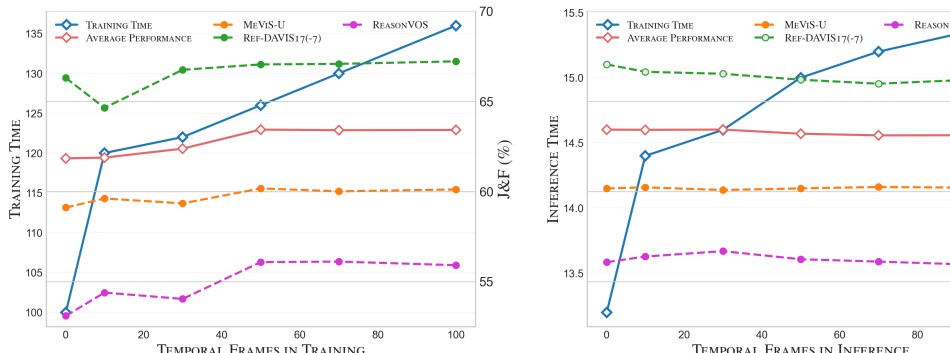

Figure 6: Effect of temporal numbers for training.      Figure 7: Effect of temporal numbers for inference.

Table 20: Effect of frame number $N_f$ on RefVOS. Evaluated with $\mathcal{J}\&\mathcal{F}$ metric.

| $N_f$ | MeViS ($val^u$) | Ref-DAVIS17 | ReasonVOS |
|---|---|---|---|
| 5 | 61.4 | 77.1 | 59.2 |
| 8 | 62.0 | 76.3 | 61.7 |
| 10 | 61.9 | 76.4 | 60.3 |
| 12 | 62.5 | 76.8 | 59.9 |
| 15 | 62.6 | 76.0 | 59.7 |

### E.3 EFFECT OF FRAME NUMBERS ON REFVOS

In the RefVOS task, the number of sampled frames strongly affects the model's ability to identify the referred target. Table 20 reports performance on three RefVOS datasets under different frame counts. On MeViS, accuracy improves as the number of frames increases, reflecting its motion-driven nature where richer temporal context aids understanding. In contrast, results on Ref-DAVIS17 and ReasonVOS fluctuate, and excessive frames can even reduce accuracy. A moderate increase in frames can help, but too many introduce irrelevant segments that degrade segmentation. Moreover, the gap between training and testing—training uses only five frames—reduces the proportion of tokenized text instructions at inference, causing the model to underutilize crucial textual cues. This mismatch warrants further investigation.

### E.4 EFFECT OF POST-PROCESSING THRESHOLD FOR TSG

For the proposed MomentSeg, the post-processing threshold $\theta$ plays a crucial role in determining the accuracy of moment localization. Table 21 demonstrates the impact of varying $\theta$ values on the performance of two TSG benchmarks, Charades-STA and ActivityNet-Grounding, evaluated using recall at different IoU thresholds (R@0.3, R@0.5, and R@0.7) and mean IoU (mIoU). We observe that for Charades-STA, increasing $\theta$ from 0.2 to 0.3 results in significant improvements across all metrics, particularly in R@0.3 and mIoU, where the best performance is achieved at $\theta = 0.3$. Further increasing $\theta$ leads to a decline in some lower-precision metrics, while higher-precision metrics continue to improve, indicating that a higher threshold favors retaining high-confidence moments, resulting in a high-precision, low-recall state. Similarly, for ActivityNet-Grounding, we observe that $\theta = 0.4$ strikes a balance between recall and precision, providing an overall stable performance across the metrics. Therefore, $\theta = 0.4$ is selected as the optimal threshold for this paper.

## F ADDITIONAL QUALITATIVE RESULTS

### F.1 VISUALIZATION COMPARISON WITH SA2VA

We compare the visual results of our proposed MomentSeg with Sa2VA on the RefVOS task. As shown in Fig. 8, three representative examples are provided. (a) Highlights a key drawback of

Table 21: Effect of post-process threshold $\theta$ on TSG.

| $\theta$ | Charades-STA | | | | ActivityNet-Grounding | | | |
|---|---|---|---|---|---|---|---|---|
| | R@0.3 | R@0.5 | R@0.7 | mIoU | R@0.3 | R@0.5 | R@0.7 | mIoU |
| 0.2 | 75.4 | 49.8 | 21.9 | 47.5 | 66.1 | 41.0 | 20.9 | 44.9 |
| 0.3 | **78.9** | 55.9 | 25.3 | **50.7** | 65.6 | **45.6** | **23.8** | 45.1 |
| 0.4 | 76.1 | **58.2** | 25.8 | 50.2 | **67.5** | 44.7 | 23.2 | **45.4** |
| 0.5 | 72.9 | 55.4 | **31.1** | 49.7 | 62.2 | 42.1 | 22.4 | 42.7 |
| 0.6 | 71.8 | 54.6 | 27.9 | 48.4 | 59.2 | 39.5 | 21.7 | 41.2 |

Sa2VA's *FirstK* sampling strategy: sampling only the early portion of a video fails to capture later object appearances, leading to missed segmentations. In contrast, our TSG module accurately localizes the action described in the text and performs dense sampling within that segment, enabling correct object segmentation. (b) Shows a case where Sa2VA entirely misses the target. The object is small and requires precise sampling; early sampling prevents Sa2VA from observing the later 'browsing phone' action. (c) Demonstrates that insufficient sampling of later key moments produces unreliable mask initialization and incomplete predictions for Sa2VA. Our method instead initializes at the most relevant frames and applies bidirectional propagation and updating, yielding more stable mask segmentation.

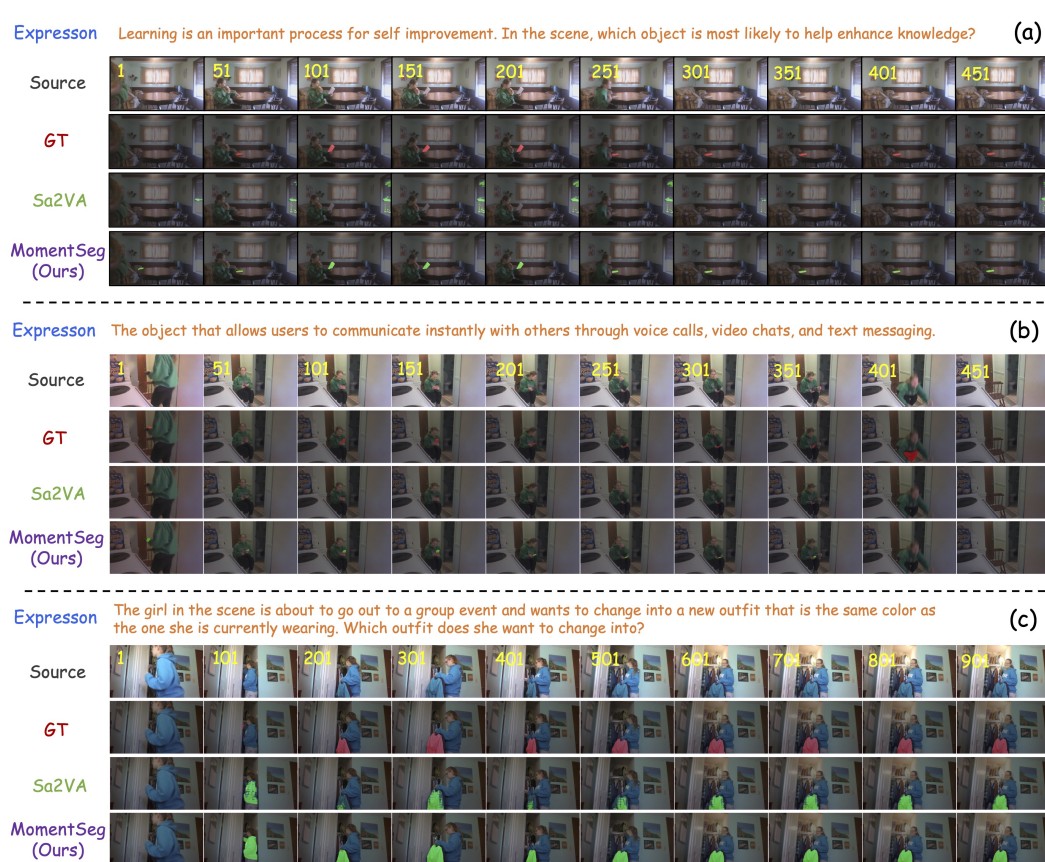

Figure 8: Additional visual comparison of MomentSeg and Sa2VA on the RefVOS task.

## F.2 REFERRING INSTRUCTION VISUALIZATION

In Fig. 9, we present additional visualization results of MomentSeg-3B on the Referring VOS task. First, it is observed that the TSG inference stage accurately localizes the action segments described by the text. Moreover, dense sampling occurs in regions with sharp peaks in the similarity distribution, as shown in the 2nd and 5th samples. This sampling strategy effectively aids the model in

understanding the action described by the text and allows for accurate tracking of the referred target during the subsequent BAP tracking process. Additionally, in complex scenarios, such as the 2nd, 3rd, and 4th samples, where multiple similar targets appear in the video, the model can still accurately segment the referred target by combining the understanding of the text description with action information.

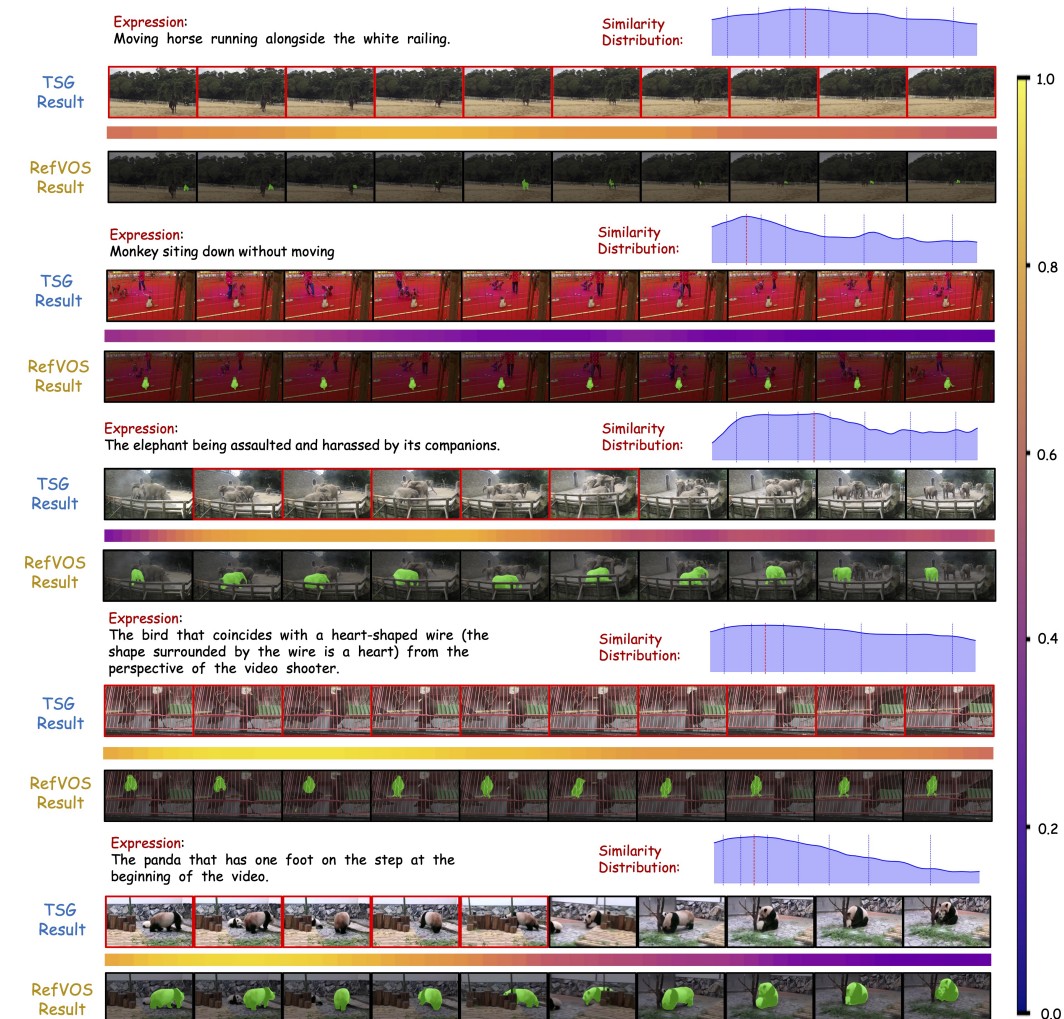

Figure 9: The additional visualization of the MomentSeg on Referring VOS task.

## F.3 REASONING INSTRUCTION VISUALIZATION

In Fig. 10, we show additional visualization results of MomentSeg-3B on the Reasoning VOS task. These examples are characterized by text descriptions that do not directly refer to a specific object but require indirect reasoning to identify the referred object. Most of these examples are in the form of questions. During this stage, it is observed that since the text does not describe much motion-related information, the TSG typically localizes the target's segment in the video rather than focusing on the critical moment. This remains a bottleneck for current methods, as we use reasoning-related TSG data for training.

## F.4 BADCASE ANALYSIS

In Fig. 11, we show some failure cases of MomentSeg-3B on the RefVOS task. (a) shows that due to incorrect localization of key segments during the TSG stage, the sampled frames are not

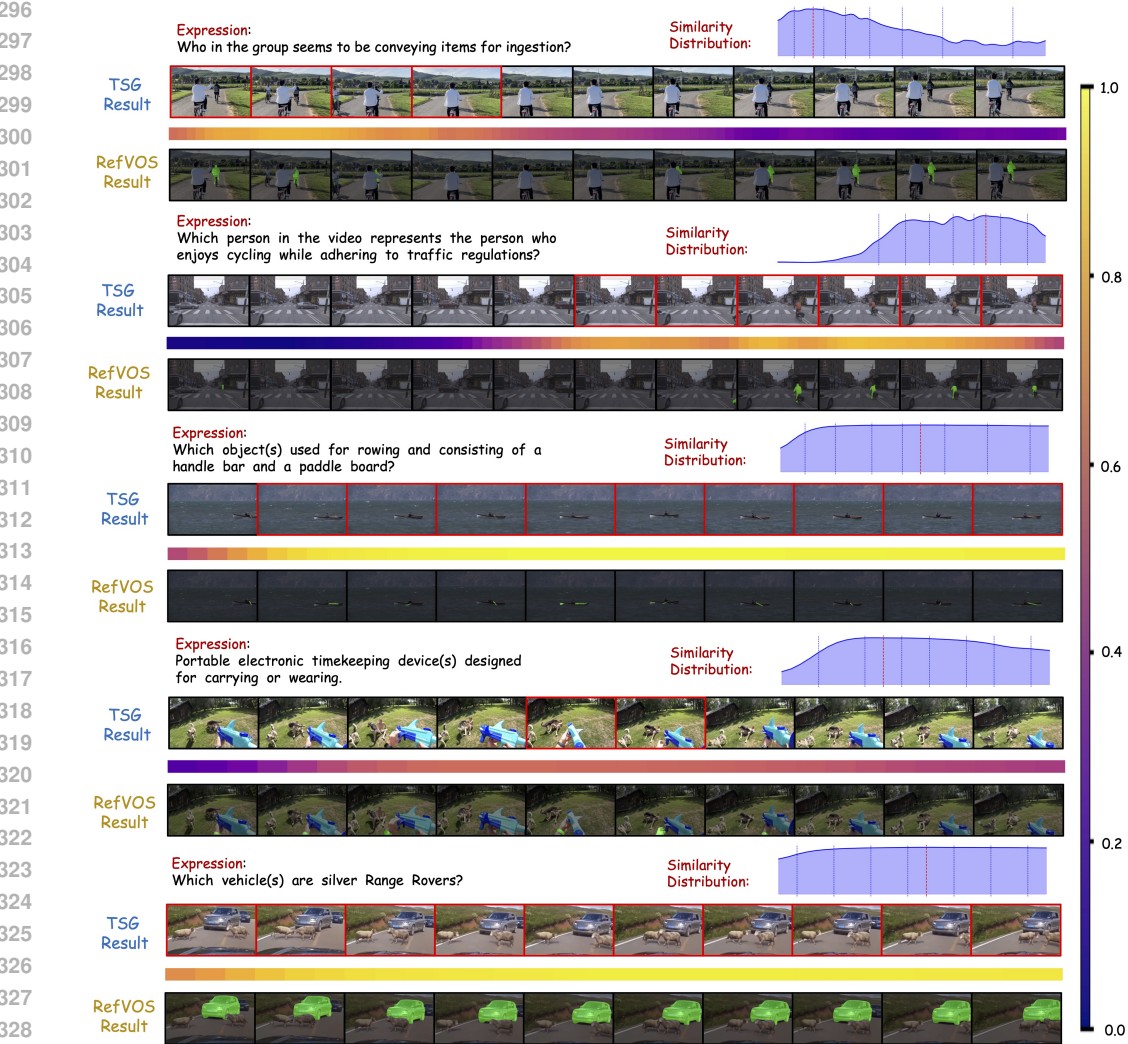

Figure 10: The additional visualization of the MomentSeg on Reasoning VOS task.

representative, and with interference from multiple targets, the model misidentifies the target as smaller, resulting in erroneous and fluctuating segmentation. (b) highlights a significant issue with current methods: the lack of encoding for frame-relative video timestamps prevents the model from understanding concepts like "at the end of the video." This leads to tracking drift and inaccurate segmentation results. (c) and (d) show cases with multiple referred objects where, due to the absence of instance-level supervision, the model performs poorly. This remains a major bottleneck in current methods.

## F.5 LIMITATIONS

MomentSeg still offers substantial room for improvement. For example, for temporal grounding, the current training data remain limited and are confined to the TSG task. Incorporating related tasks—such as Dense Video Captioning (Iashin & Rahtu, 2020), Video Highlight Detection (Lei et al., 2021), and Temporally Grounded Video Question Answering (Chen et al., 2024a; Liu et al., 2024c)—could enhance the model's generalization ability. Moreover, the number of sampled frames is currently fixed; future work could explore dynamically adapting the sampling rate to different video complexities, allowing more frames to assist understanding in scenes with intricate descriptions and thereby enabling more accurate target segmentation.

### F.6 FUTURE WORK

Certain video-related tasks, such as **Temporal Action Segmentation (TAS)** (Fathi et al., 2011; Kuehne et al., 2014), require dense temporal annotations. These tasks share a conceptual connection with MomentSeg's TSG module, suggesting that methods developed in the TAS domain (Li et al., 2020; Farha & Gall, 2019; Yi et al., 2021) could be adapted to enhance MomentSeg's temporal segmentation capabilities.

Furthermore, extending MomentSeg to **streaming video scenarios**, as explored in VideoLLM-Online (Chen et al., 2024b), represents another promising research direction. In particular, designing mechanisms analogous to the **[EOS] token** to enable efficient frame-by-frame online segmentation could further broaden MomentSeg's applicability to real-time video processing tasks.

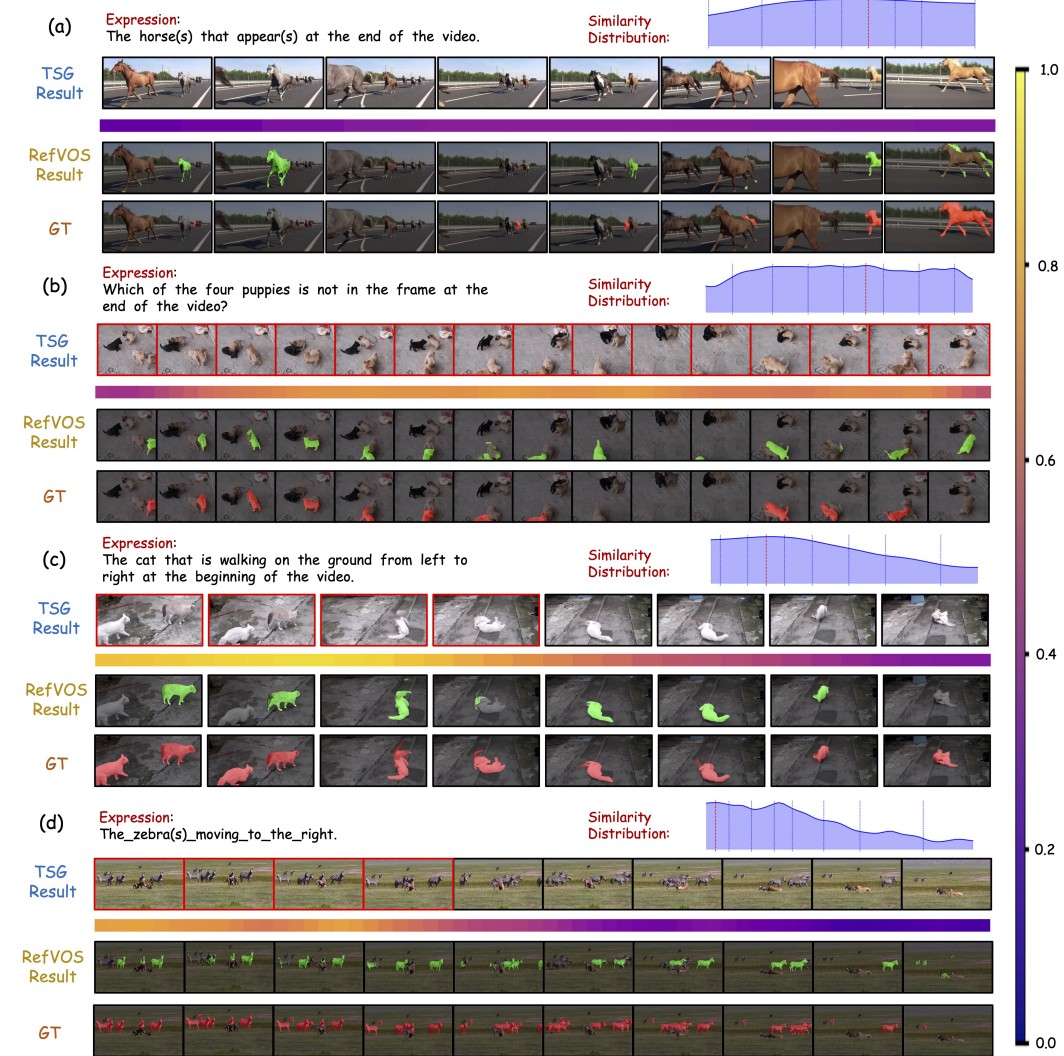

Figure 11: Additional visualization of MomentSeg failure cases.

