# OpenReview forum: "MomentSeg: Moment-Centric Sampling for Enhanced Video Pixel Understanding"
_ICLR.cc/2026/Conference — Submitted to ICLR 2026_

### Official Review · Reviewer_zWDN · 2025-10-25

**Soundness:** 3
**Presentation:** 3
**Contribution:** 4
**Rating:** 6
**Confidence:** 4

**Summary:**

This thesis addresses the shortcomings of existing large language model (LMM)-based approaches in the referential video target segmentation (RefVOS) task i.e., either relying on external keyframe models to increase the system complexity or adopting a manual sampling strategy to lose the critical timing cues i.e., proposes a unified framework MomentSeg to achieve the joint optimization of temporal sentence localization (TSG) and RefVOS to improve the video pixel-level comprehension. In the training phase, [FIND] token is introduced to identify text-related key frames by frame-level temporal tokn similarity matching without external timestamp encoding; in the inference phase, a moment-centered sampling (MCS) strategy is designed to densely sample high-relevance key moments and sparsely sample non-key frames, and a bi-directional anchor update propagation (BAP) is proposed to bi-directionally propagate the forward and backward temporal sequence from key frames At the same time, we propose the bidirectional anchor update propagation (BAP), which propagates the mask from the key frames to the forward and backward time sequences in a bi-directional manner and updates it adaptively to reduce the tracking accumulation error.
Its core contributions include 1. Proposing MomentSeg unified framework, which realizes native keyframe selection by matching the similarity between [FIND] token and frame token during training, which eliminates the need of relying on external keyframe models and the need of explicit timestamp encoding, and simplifies the system design. 2. Design Moment Centered Sampling (MCS): Based on the similarity distribution of [FIND]tokn output, dense sampling at text-related key moments and sparse sampling in non-critical regions, taking into account the preservation of key motion details and global timing context maintenance. 3. Bidirectional Anchor Update Propagation (BAP) is proposed: taking the key frames recognized by MCS as the starting point, the mask is propagated forward and backward chronologically in a bidirectional manner, and the mask and memory are updated adaptively at the sampling points, which can effectively alleviate the cumulative tracking error and improve the robustness of long video segmentation.

**Strengths:**

1. The moment-centric modeling formulation is conceptually sound and backed by a clear motivation:local semantic coherence is often more stable than frame-wise predictions.
2. The appendix adds significant methodological rigor,defining how temporal tokens and attention masks are constructed,and validating their influence.
3. The ablation studies are systematic and reproducible-exploring token number,temporal frame count,threshold sensitivity,and alternative output paradigms ([FIND]vs.text generation).

**Weaknesses:**

1. The moment definition remains implicit:moments are derived implicitly from attention maps,not through an explicit segmentation constraint This makes reproducibility somewhat architecture-dependent.
2. The causal or theoretical justification for why "moment aggregation"improves generalization is intuitive but lacks formal backing (e.g.,no information-theoretic analysis or invariance claim).
3. Temporal boundary precision is still limited;CRF helps smooth predictions but does not guarantee sharp semantic transition detection.
4. Efficiency considerations are largely omitted (e.g.,FLOPs,latency),though essential for TAS scalability.

**Questions:**

1. Efficiency Analysis Needed: No report of inference time or computational overhead-important for real-world deployment.
2. Comparative Positioning vs.Prior Work: The relation to window-based methods (MS-TCN++,ASFormer)should be analytically clarified-e.g.,how MomentSeg differs from dynamic temporal pooling or multi-scale convolution.
3. Generalization Study Beyond Benchmarks: ncluding transfer experiments (e.g.,from GTEA>Breakfast)could demonstrate robustness beyond in-distribution evaluations.

---

> ### Author Response · Authors · 2025-11-19
> **Reponse to Reviewer zWDN (Q1)**
>
> We sincerely thank the reviewer for the positive evaluation and the constructive suggestions, such as those regarding the **Efficiency Analysis**. We attempt to address the reviewer’s concerns below based on our current understanding; if any of our interpretations are inaccurate, we would greatly appreciate further clarification.
>
> In **Global Comment G1**, we have summarized the core contributions of **MomentSeg** to provide a clear overview of its main innovations.
>
> **Q1:** The moment definition remains implicit: moments are derived implicitly from attention maps, not through an explicit segmentation constraint. This makes reproducibility somewhat architecture-dependent.
>
> **A1:** We thank the reviewer for raising this point. In fact, the keyframe (moment) localization in **MomentSeg does *not* rely on attention maps**. Instead, the [FIND] token serves as a sentence-aware implicit embedding. During inference, the [FIND] token computes its similarity with frame-level pooled embeddings, producing a sentence-conditioned **frame-level similarity distribution**. This similarity curve defines the temporal moments implicitly, without depending on any specific attention pattern or architectural detail.
>
> During training, the TSG and RefVOS datasets are kept independent. However, we adopt a **multi-task joint training strategy**, where different input formats and prompts supervise different tasks within the same unified architecture. At inference time, the model first performs TSG to generate the similarity distribution, and then the MCS module transfers this distribution to the RefSeg task for consistent temporal grounding.

---

> ### Author Response · Authors · 2025-11-19
> **Reponse to Reviewer zWDN (Q2)**
>
> **Q2:** The causal or theoretical justification for why "moment aggregation"improves generalization is intuitive but lacks formal backing (e.g.,no information-theoretic analysis or invariance claim).
>
>
> **A2:** We respectfully clarify that our paper neither mentions nor involves the concept referred to as “moment aggregation.” We would appreciate further clarification from the reviewer to better understand and address their concern.
>
> To enhance the clarity of our method, we have revised the presentation of **Algorithm 1** to more precisely describe the operations of MCS. Additionally, we have strengthened the experimental discussion of MCS in **Section 4.3** ("**Effectiveness of Moment-Centric Sampling**").
>
> Moreover, we provide an in-depth analysis of the robustness of BAP, including improvements in both initialization and update procedures. The corresponding results are reported in **Table 13** and further detailed in the "**Analysis of Robustness**" subsection in **Section 4.3**.

---

> ### Author Response · Authors · 2025-11-19
> **Reponse to Reviewer zWDN (Q3)**
>
> **Q3:** Temporal boundary precision is still limited; CRF helps smooth predictions but does not guarantee sharp semantic transition detection.
>
>
> **A3:** We thank the reviewer for the constructive comment. We acknowledge that computing frame-level similarity scores based on the [FIND] token inherently restricts the temporal boundary resolution to the frame level. To mitigate the impact of limited input frames, we apply bilinear interpolation to diffuse the similarity distribution across all video frames as an approximation.
>
> A straightforward way to further improve boundary accuracy would be to feed more temporal frames into the model; however, this inevitably increases computational overhead and contradicts our design goal of maintaining efficiency.
>
> Regarding the use of CRF, we do not adopt CRF-based refinement since our model outputs only a similarity distribution rather than discrete per-frame labels. Instead, we apply a lightweight Gaussian smoothing to stabilize the similarity curve without incurring additional inference complexity.

---

> ### Author Response · Authors · 2025-11-19
> **Reponse to Reviewer zWDN (Q4)**
>
> **Q4** Efficiency Analysis Needed: No report of inference time or computational overhead-important for real-world deployment.
>
> **A4:** We have provided a detailed analysis of the inference efficiency of each component in **Table 12** and **Section 4.3 ("Analysis of Inference Latency")** in the main text. (Please also see our **Global Comment G2** for a summary).
> In this analysis, we compare MomentSeg directly to Sa2VA, which uses a simpler, single-pass sampling strategy. The results show that our method achieves substantial performance improvements on RefVOS (**+10.2% on MeViS** and **+7.5% on ReVOS**) while **not adding significant inference latency (3%)**.
>
>
> | Method | Ref-Youtube-VOS (J&F) | Ref-Youtube-VOS (Latency (s)) | MeVIS (val) (J&F) | MeVIS (val) (Latency (s)) | ReVOS (J&F) | ReVOS (Latency (s)) |
> | :--- | :---: | :---: | :---: | :---: | :---: | :---: |
> | Sa2VA-8B | 70.7 | 6.8 | 46.9 | 7.5 | 57.6 | 6.3 |
> | **MomentSeg-7B (Ours)** | 72.3(↑1.6) | 7.0(↓3%) | 57.1(↑10.2) | 7.7(↓3%) | 65.1(↑7.5) | 6.4(↓2%) |
> | ↳ Preprocess | - | 1.8 (26%) | - | 3.3 (43%) | - | 2.4 (38%) |
> | ↳ MLLM | - | 1.3 (19%) | - | 1.4 (18%) | - | 1.3 (20%) |
> | **&emsp;**↳ TSG | - | 0.5 (7%) | - | 0.6 (8%) | - | 0.5 (8%) |
> | **&emsp;**↳ RefVOS | - | 0.8 (12%) | - | 0.8 (10%) | - | 0.8 (12%) |
> | ↳ SAM2 | - | 3.9 (55%) | - | 3.0 (39%) | - | 2.7 (42%) |

---

> ### Author Response · Authors · 2025-11-19
> **Reponse to Reviewer zWDN (Q5)**
>
> **Q5:** Comparative Positioning vs.Prior Work: The relation to window-based methods (MS-TCN++,ASFormer)should be analytically clarified-e.g.,how MomentSeg differs from dynamic temporal pooling or multi-scale convolution.
>
> **A5:** We thank the reviewer for this insightful question, which helps us clarify our positioning.
>
> The methods mentioned (MS-TCN++[1], ASFormer[2]) are typically designed for Temporal Action Segmentation (TAS), often processing video features through multi-scale or windowed attention/convolution mechanisms to find *all* action boundaries, independent of language.
>
> Our approach is fundamentally different as it is **Auto-regressive** method for the TSG, RefVOS, ImageQA, VideoQA and RefSeg tasks.
>
> * **Mechanism Distinction:** Instead of using fixed windows or multi-scale convolutions to aggregate features, our method employs a `[FIND]` token. This token learns to represent the *semantics of the text query*.
> * **Dynamic vs. Fixed Pooling:** The "pooling" or "selection" in our method is not a pre-defined window operation. It is a **dynamic, similarity-driven process**. The `[FIND]` token computes a relevance score against *all* temporal frames, generating a *frame-query similarity distribution*.
> * **Query-Conditioned Sampling:** This distribution, which is unique and specific to each query, is then used by our Moment-Centric Sampling (MCS) to select frames.
>
> In short, window-based methods aggregate features based on modeling temporal structure, whereas our method selects frames based on **semantic alignment** between a specific language query and the video content.
>
> However, we acknowledge that TAS is also a highly relevant research direction and offers promising opportunities for future extensions. In response to the reviewer’s suggestion, we have added a discussion of MS-TCN++, ASFormer, and related approaches in the future work section, highlighting their relevance and how such techniques could inspire potential extensions of our framework. We discussed about this in the **Section F.6** in the revised version.
>
> [1] MS-TCN++: Multi-Stage Temporal Convolutional Network for Action Segmentation, TPAMI2020
>
> [2] ASFormer: Transformer for Action Segmentation, BMVC2021

---

> ### Author Response · Authors · 2025-11-19
> **Reponse to Reviewer zWDN (Q6)**
>
> **Q6:** Generalization Study Beyond Benchmarks: including transfer experiments (e.g.,from GTEA>Breakfast)could demonstrate robustness beyond in-distribution evaluations.
>
> **A6:** TAS is indeed an important task. However, our paper is fundamentally centered on segmentation. TSG is introduced to provide sentence-aware keyframes that better support the segmentation process, whereas TAS requires frame-level classification across the entire video.
>
> Our current work primarily centers on language-conditioned video understanding tasks such as Referring Video Object Segmentation (RefVOS), Temporal Sentence Grounding (TSG), and Video Chat, where natural language is the core input modality. Evaluating the model's transferability on temporal action segmentation benchmarks like GTEA and Breakfast would necessitate substantial modifications to the supervision scheme and cannot be achieved via straightforward transfer. Furthermore, our MomentSeg framework is also adapted for image tasks like Referring Image Segmentation (RefSeg) and Image Question Answering (ImageQA), with results reported across more than 20 datasets in the paper.
>
> While this lies beyond the scope of our current study, we fully agree that it represents a valuable direction for future research. Accordingly, we have added this point to the **Future Work (Section F.6)** in the revised manuscript.

---

> ### Comment · Area_Chair_1DmS · 2025-11-25
>
> Dear Reviewer zWDN,
>
> The authors have responded to your reviews. Please review and provide your feedback and responses.
>
> Best,
>
> Your AC

---

> ### Comment · Reviewer_zWDN · 2025-11-25
>
> I appreciate authors' reply. Honestly, this is a good job.
>
> After reviewing all authors' comments, I decide to keep my original score.

---

### Official Review · Reviewer_62dL · 2025-10-30

**Soundness:** 3
**Presentation:** 4
**Contribution:** 2
**Rating:** 4
**Confidence:** 3

**Summary:**

## Summary
This paper proposes MomentSeg for joint TSG and RefVOS training with [FIND] token-based keyframe selection, moment-centric sampling (MCS), and bidirectional anchor-updated propagation (BAP). While empirical results show improvements, the work suffers from limited novelty, questionable experimental design, and theoretical gaps.

**Strengths:**

## Strengths

1. **Comprehensive empirical evaluation**: The paper provides extensive experiments across multiple benchmarks including TSG (Charades-STA, ActivityNet), RefVOS (MeVIS, ReVOS, Ref-Youtube-VOS, Ref-DAVIS17), image segmentation (RefCOCO/+/g, ReasonSeg, GCG), and QA tasks. This breadth demonstrates the method's versatility.

2. **Consistent improvements across benchmarks**: The method shows meaningful gains over strong baselines, particularly +5% on MeVIS valu (62.0 vs 57.5) and +6% on ReVOS overall (62.6 vs 58.7) compared to Sa2VA baseline. The 3B model outperforms several 7B models on certain tasks.

3. **Thorough ablation studies**: Tables 6-10 provide detailed ablations of key components (TTI, MCS, BAP), different sampling strategies (Table 7), and joint training effects (Table 10). The analysis of sampling strategies in Fig. 2 effectively motivates the problem.

4. **Eliminates external keyframe selection models**: Unlike VISA, ViLLA, GLUS, and VRS-HQ which require separate models (LLaMA-VID, Grounded-VideoLLM, CLIP) for keyframe identification, this method learns keyframe selection end-to-end within the main model, reducing pipeline complexity.

5. **Well-motivated design**: The analysis showing that random sampling has high variance (Fig. 2) and that uniform sampling outperforms firstK on motion-driven datasets provides good empirical motivation for learned keyframe selection.

6. **Strong TSG results**: Table 2 shows impressive performance on temporal grounding (76.1 R@0.3, 58.2 R@0.5 on Charades-STA) despite using a smaller 3B model, outperforming methods like VTimeLLM-7B and HawkEye-7B.

7. **Detailed supplementary material**: The 16-page appendix includes additional experiments, visualizations, implementation details, and failure case analysis, demonstrating thoroughness.

8. **Clear presentation**: The paper is generally well-written with informative figures (especially Fig. 1, 3, 4) that effectively illustrate the approach. The distinction between training and inference pipelines is clearly explained.

**Weaknesses:**

### Concerns Regarding the Novelty of Proposed Components

While the paper achieves impressive empirical results, I have some concerns regarding the claimed novelty of the core technical components ([FIND] token, MCS, and BAP), which appear to be effective applications of well-established principles.

**1. On the [FIND] Token Paradigm**

The paper introduces the [FIND] token as part of a "novel TSG paradigm". However, the described mechanism—a learnable token that is matched against frame features using cosine similarity and trained with a contrastive loss—bears a strong resemblance to standard learnable queries used in attention-based models (e.g., DETR) and has been widely explored in video grounding literature.

* **Request for Clarification:** Could the authors elaborate on the specific technical distinctions that qualify this as a "novel paradigm" rather than a successful and well-executed application of existing query-key matching mechanisms for the task of temporal grounding?

**2. On Moment-Centric Sampling (MCS)**

The proposed MCS strategy samples frames based on the similarity distribution generated by the [FIND] token, effectively sampling more densely from regions of high relevance.

* **Observation:** This method appears to be a direct application of importance sampling. The use of Inverse CDF sampling, as detailed in Algorithm 1, is a textbook statistical method for drawing samples from an arbitrary probability distribution. While this is a sensible and effective choice, it may be an overstatement to present the sampling algorithm itself as a novel contribution.

**3. On Bidirectional Anchor-updated Propagation (BAP)**

The BAP mechanism is presented as a method to enhance tracking robustness.

* **Observation:** The key ideas of (1) initializing a tracker from a high-confidence "anchor" frame and (2) propagating both forward and backward in time are standard and widely-used techniques in the video object segmentation (VOS) field to prevent error accumulation.
* **Request for Clarification:** The ablation in Table 8 also shows that the full BAP (with Bidirectional Propagation) results in a slight performance drop on Ref-DAVIS17 (76.8 vs. 77.0) compared to just using "Moment-anchored Updating." This seems to slightly undermine the claim of its universal benefit. Could the authors clarify the specific novelty of BAP beyond combining these existing VOS techniques, and perhaps comment on this performance discrepancy?

**Summary of Concern:**

The primary contribution of this work appears to be the skillful combination and joint optimization of these components into a high-performing, unified framework for RefVOS and TSG. While the engineering is strong and the results are state-of-the-art, the paper would be more impactful if it more clearly distinguished its fundamental research contributions from the novel application of existing, well-established methods.

**Questions:**

1. Inference Cost
    - Could you provide a detailed breakdown of the inference latency and/or FLOPs for the full MomentSeg pipeline? How does it compare to other LMM-based methods that might use a simpler, single-pass sampling strategy?

---

> ### Author Response · Authors · 2025-11-19
> **Reponse to Reviewer 62dL (Q1)**
>
> We thank you for your careful analysis of our paper. We have provided detailed explanations and analyses for the clarifications you requested, and we hope these satisfy your requirements for the paper. If anything remains unclear, we welcome any further questions. We have also summarized our main contributions in Global Comment **G1**.
>
> **Q1:** Could the authors elaborate on the specific technical distinctions that qualify this as a "novel paradigm" rather than a successful and well-executed application of existing query-key matching mechanisms for the task of temporal grounding?
>
> **A1:** The `[FIND]` token strategy differs from existing methods.
>
> * First, **DETR-based** methods (like Moment-DETR[1], QD-DETR[2]), which use pre-defined learnable "moment queries" that are decoded with video embeddings to predict moment boundaries.
> * Second, **LLM-based** methods (like VTG-LLM[3], NumberIT[4]), as discussed in our Related Works (**Lines 173-183**), often use a plain-text generation paradigm. A key problem there is that the model must be explicitly told which frame it is processing, which requires **timestamps encoding**.
>
> The core of our design is that the `[FIND]` token computes a similarity distribution directly with the temporal frame features. This **circumvents the need for extra explicit timestamp encoding**. While traditional LLM-based TSG might use instruction tuning to generate plain-text outputs like `[start timestamp, end timestamp]`, our `[FIND]` token serves as an **implicit representation** of a temporal interval, much like the `[SEG]` token serves as an implicit representation of a mask.
>
> [1] QVHighlights: Detecting Moments and Highlights in Videos via Natural Language Queries, NeurIPS 2021
>
> [2] QD-DETR : Query-Dependent Video Representation for Moment Retrieval and Highlight Detection, CVPR 2023
>
> [3] Vtg-llm: Integratingtimestamp knowledge into video llms for enhanced video temporal grounding. AAAI2025
>
> [4] Number it: Temporal grounding videos like flipping manga. CVPR2025

---

> ### Author Response · Authors · 2025-11-19
> **Reponse to Reviewer 62dL (Q2)**
>
> **Q2:** The proposed MCS strategy samples frames based on the similarity distribution generated by the [FIND] token, effectively sampling more densely from regions of high relevance. This method appears to be a direct application of importance sampling. The use of Inverse CDF sampling, as detailed in Algorithm 1, is a textbook statistical method for drawing samples from an arbitrary probability distribution. While this is a sensible and effective choice, it may be an overstatement to present the sampling algorithm itself as a novel contribution.
>
> **A2:**
> We thank the reviewer for carefully analyzing the details of MCS. As you correctly point out, MCS is indeed a strategy for keyframe sampling based on the similarity distribution generated by the [FIND] token, and Inverse CDF sampling is the well-established tool we use to implement it.
>
> We did not intend to claim the sampling algorithm itself as a novel statistical method. Rather, the contribution lies in the **design and specific application** of this sampling within our framework:
>
> 1. The sampling is **enabled by our [FIND] token's design**, which first generates the similarity distribution. This is the key prerequisite, shifting the task from generating fixed plain text (e.g., timestamps) to localizing via a similarity-based soft distribution.
> 2. MCS uses this distribution to perform two specific functions: first, it identifies the single **most relevant keyframe**; second, it performs **dense sampling in this text-relevant region** while **sparsely sampling irrelevant areas**. This preserves both critical motion details and global context, as motivated in our Introduction.
> 3. The key temporal points sampled by MCS are then used as the **anchor nodes for our Bidirectional Anchor-updated Propagation (BAP)**, enabling mask updates and error correction.

---

> ### Author Response · Authors · 2025-11-19
> **Reponse to Reviewer 62dL (Q3)**
>
> **Q3:** The ablation in Table 8 also shows that the full BAP (with Bidirectional Propagation) results in a slight performance drop on Ref-DAVIS17 (76.8 vs. 77.0) compared to just using "Moment-anchored Updating." This seems to slightly undermine the * claim of its universal benefit. Could the authors clarify the specific novelty of BAP beyond combining these existing VOS techniques, and perhaps comment on this performance discrepancy?
>
> **A3:**
> We thank the reviewer for this detailed observation. The question involves two aspects: (1) the performance discrepancy on Ref-DAVIS17, and (2) the novelty of BAP. We address them separately below.
>
> 1. **On the Ref-DAVIS17 Performance:**
>    The three datasets differ significantly in their characteristics. MeVIS includes many motion-centric descriptions and long videos (>100 frames), while ReasonVOS contains even longer videos (>300 frames) and requires strong video–text reasoning. In contrast, **Ref-DAVIS17 consists of short videos where the target is present in almost every frame, accompanied by simple textual descriptions.**
>    Our *Bidirectional Propagation* module is specifically designed for *long sequences*, where identifying the key moment is essential for stable mask initialization. When the target is always visible and easy to localize—as in Ref-DAVIS17—this specialized initialization strategy offers little benefit. Thus, the slight fluctuation (76.8 vs. 77.0) is negligible. As shown in Table 9, the advantage of this component becomes evident in long and complex datasets such as **MeVIS and ReasonVOS**, where it provides a **+0.7% improvement** by ensuring propagation begins at the most text-relevant moment.
> 2. **On the Novelty of BAP:**
>    BAP introduces several targeted enhancements to the SAM2 propagation pipeline. Its novelty stems from three new components designed to function cohesively with our MCS module:
>
>    * **(1) Moment-anchored Updating:** The MCS-selected keyframes serve as “anchors” to **actively correct masks** during propagation. This prevents errors from simply propagating through the sequence. As Table 9 shows, this single component yields significant gains (**+2.2% on MeVIS and +3.4% on ReasonVOS**).
>    * **(2) Adaptive Memory Cleaning:** This mechanism dynamically clears the memory to mitigate error accumulation over long sequences. It contributes a small but consistent improvement across all datasets.
>    * **(3) Bidirectional Propagation:** Using the most text-relevant frame predicted by MCS as the **starting point**, we initialize the mask in a frame where the target is most likely to appear. This avoids the common failure of forward-only propagation, which may initialize from an empty target frame. This component adds another **+0.7%** improvement on long, complex scenarios (MeVIS and ReasonVOS).
>
> Additionally, we have added a more extensive robustness analysis of BAP in Table 13 and Section 4.3 (*Analysis of Robustness*) of the main paper. For further details, please refer to **A5 for Reviewer CUfj**.

---

> ### Author Response · Authors · 2025-11-19
> **Reponse to Reviewer 62dL (Q4)**
>
> **Q4:** The primary contribution of this work appears to be the skillful combination and joint optimization of these components into a high-performing, unified framework for RefVOS and TSG. While the engineering is strong and the results are state-of-the-art, the paper would be more impactful if it more clearly distinguished its fundamental research contributions from the novel application of existing, well-established methods.
>
> **A4:** We thank the reviewer for this important feedback. We have summarized our main contributions in **Global Comment G1**.
>
> To more clearly distinguish our fundamental contributions from novel applications, we highlight the following core differences from prior work:
>
> * We jointly train two special tokens, `[FIND]` and `[SEG]`, to unify the TSG and RVOS tasks. This is not just multi-tasking; the TSG task is designed to **directly serve the RefVOS task** (by providing the similarity distribution for sampling). We also found that this joint optimization **mutually improves both tasks** (as shown in Table 11, TSG improves by +1.2% and ReVOS by +0.9%).
> * For the TSG task, our `[FIND]` token's similarity-based matching mechanism **eliminates the need for explicit timestamp encoding**, which is a required, complex design in prior methods (like Grounded-VideoLLM or VTG-LLM). Furthermore, it generates a frame-level relevance score, which, unlike plain-text generation (e.g., `[start, end]`), offers **greater flexibility**. For example, users can dynamically adjust the post-processing threshold to trade off precision and recall for different applications.
> * We analyzed the importance of keyframes and proposed **Moment-Centric Sampling (MCS)**. This is not a simple application of sampling; it is a **similarity-driven strategy** that leverages the `[FIND]` token's output to perform **dense sampling in text-relevant segments (for local detail) and sparse sampling elsewhere (for global context)**. This provides a more reliable `[SEG]` embedding for the RefVOS task (as shown in **Table 7**, introducing MCS improves MeVIS by +2.6 points).
> * We proposed **Bidirectional Anchor-updated Propagation (BAP)**. This is a new propagation mechanism that (1) **initializes from the key moment** identified by MCS (not the start of the video), (2) propagates bidirectionally, and (3) **dynamically updates and clears memory** at the sampled anchor points. This mechanism is designed to **actively correct accumulated errors**, which is a key weakness in standard forward propagation (as shown in **Table 7**, BAP contributes a +2.2 point gain on ReVOS).

---

> ### Author Response · Authors · 2025-11-19
> **Reponse to Reviewer 62dL (Q5)**
>
> **Q5:** Could you provide a detailed breakdown of the inference latency and/or FLOPs for the full MomentSeg pipeline? How does it compare to other LMM-based methods that might use a simpler, single-pass sampling strategy?
>
> **A5:** We thank you for this suggestion. We have provided a detailed analysis of the inference efficiency of each component in **Table 12** and **Section 4.3 ("Analysis of Inference Latency")** in the main text. (Please also see our **Global Comment G2** for a summary).
> In this analysis, we compare MomentSeg directly to Sa2VA, which uses a simpler, single-pass sampling strategy. The results show that our method achieves substantial performance improvements on RefVOS (**+10.2% on MeViS** and **+7.5% on ReVOS**) while **not adding significant inference latency (3%)**.

---

> ### Comment · Area_Chair_1DmS · 2025-11-25
>
> Dear Reviewer 62dL,
>
> The authors have responded to your reviews. Please review and provide your feedback and responses.
>
> Best,
>
> Your AC

---

> ### Author Response · Authors · 2025-11-27
> **Comment by authors**
>
> Dear Reviewer,
>
> Thank you again for your time and valuable suggestions.
>
> As the rebuttal period is approaching its end with **less than one week remaining**, we wanted to kindly check if there are any remaining concerns or questions that we could help clarify.
>
> If you have **any additional points or technical questions**, please feel free to raise them — we would try our best to provide further explanation or experiments to support a thorough evaluation.
>
> We sincerely appreciate your insights and look forward to your reply.
>
> Best regards,
> Authors of Paper 1310

---

### Official Review · Reviewer_CUfj · 2025-10-30

**Soundness:** 2
**Presentation:** 3
**Contribution:** 2
**Rating:** 4
**Confidence:** 4

**Summary:**

This paper proposes MomentSeg, a novel framework for referring video object segmentation (RVOS) that addresses the limitations of sparse and fixed sampling in existing methods. The core contribution is the Moment-Centric Sampling (MCS) mechanism, which intelligently selects frames most relevant to the referring expression to improve temporal grounding and visual context. Furthermore, the model employs a Bidirectional Anchor-updated Propagation (BAP) strategy to ensure robust and temporally consistent mask tracking across the selected video segment. The experimental results, validated across multiple challenging datasets, demonstrate that MomentSeg achieves state-of-the-art or highly competitive performance, confirming the effectiveness of the proposed sampling and propagation strategies.

**Strengths:**

1. Clear Motivation: The work provides a clear and compelling motivation, effectively highlighting the inherent limitations of conventional fixed or simple keyframe sampling approaches when dealing with moment-centric video-language tasks.
2. Comprehensive Experiments and Strong Results: The experimental evaluation is notably comprehensive, providing ample technical detail and achieving excellent, state-of-the-art results across a diverse set of benchmark datasets.
3. Well-Structured and Accessible Methodology: The overall methodology and presentation of the paper are clear, logical, and easy to follow, making the proposed MomentSeg architecture and techniques accessible to the broader reader community.

**Weaknesses:**

1. Incremental Gain of MCS: The core idea of "keyframe mining" appears somewhat incremental, and the reported empirical gains are small; Figure 2 suggests the impact of the sampling strategy is marginal (around 2 points), and Table 7 confirms that the gain provided by MCS over the base keyframe methods is minimal.
2. Similarity to Prior Grounding Work: The approach of using the [FIND] token to localize keyframes is highly similar to the established "streaming EOS prediction" mechanism found in related work, such as VideoLLM-Online, which diminishes the technical novelty of this specific component.
3. Limited Applicability to Online Scenarios: Key components of the proposed method, including Moment-Centric Sampling (MCS) and Bidirectional Anchor-updated Propagation (BAP), fundamentally require access to the entire video's context, severely limiting their applicability to real-world online or streaming environments.
4. Novelty of Bidirectional Propagation: While effective within this framework, the concept of bidirectional propagation for ensuring temporal consistency is not a novel technique in the broader Video Object Segmentation (VOS) literature and represents mainly an adaptation to the SAM model rather than a core innovation.
5. Missing Robustness Analysis: The paper lacks a deep-dive analysis on the method's robustness, particularly how BAP performs when anchor frames contain significant noise or the model encounters high degrees of occlusion, which are common challenges in propagation-based models.

[*] VideoLLM-online: Online Video Large Language Model for Streaming Video, CVPR2024.

**Questions:**

See the weakness.

---

> ### Author Response · Authors · 2025-11-19
> **Reponse to Reviewer CUfj (Q1)**
>
> **Q1:** Incremental Gain of MCS: The core idea of "keyframe mining" appears somewhat incremental, and the reported empirical gains are small; Figure 2 suggests the impact of the sampling strategy is marginal (around 2 points), and Table 7 confirms that the gain provided by MCS over the base keyframe methods is minimal.
>
> **A1:** We thank the reviewer for their observation and would like to clarify the impact of Moment-Centric Sampling (MCS) and the context of our experimental results.
>
> * **Gains over Baseline (Table 7 now Table 8):** We first want to clarify the comparison to the 'FirstK' sampling baseline (which mimics the strategy of Sa2VA). As shown in **Table 8**, our MCS provides a substantial improvement over 'FirstK', achieving **+3.0% on MeViS ($val^u$)** and a **+4.9% on ReasonVOS**. We believe these gains are significant.
> * **Context for Keyframe Methods (Table 8):** Regarding the comparison methods presented in Table 8 ("KeyFrame (TopK)" and "KeyFrame (NearbyK)"), it is crucial to note that these ablations utilize the same initial similarity scores derived from our TSG-trained [FIND] token design. MCS is fundamentally a superior and lightweight (CPU-only) sampling logic operating on these identical scores. Although the performance gain of MCS over the 'TopK' baseline is modest (e.g., a $+0.4\%$ improvement on MeViS and $+0.9\%$ on ReasonVOS), MCS consistently emerges as the best-performing strategy while requiring minimal computational overhead. For clarity, we have revised the main paper (*Line 280*) and **Section 4.3**, "*Effectiveness of Moment-Centric Sampling*," to incorporate this necessary context.
> * **Context for Figure 2:** We must also clarify that **Figure 2** was part of an *initial, motivational study* conducted before we developed the final MomentSeg framework. It was trained on partial data to validate our initial hypothesis that "keyframe selection is necessary." Therefore, the results in this figure should be seen as a starting point, and the absolute scores are not reflective of our final model. However, we note that the **overall trend is almost consistent** with our final results: the sampling strategy provides significant gains on datasets requiring complex temporal understanding, such as **MeViS** (action-driven descriptions) and **ReasonVOS** (reasoning in long videos), while the improvement is smaller on datasets like **Ref-DAVIS** (short videos, simple descriptions). This early finding was a key motivation for our investigation.
> * **Overall Model Performance:** The true contribution of our framework (TSG training + MCS sampling) is best seen in the end-to-end performance on challenging temporal datasets, where the gains are substantial: (1) Our MomentSeg-3B model outperforms the **Sa2VA-4B** baseline by **+9.4%** on ReVOS (62.6 vs. 53.2) and **+8.6%** on MeVIS (val) (54.8 vs. 46.2). (2) When controlling for the MLLM (using the same Qwen2.5-VL-3B backbone), our MomentSeg-3B model outperforms the Sa2VA-Qwen2.5-VL-3B baseline by **+12.1%** on ReasonVOS (61.7 vs. 49.6) and **+4.8%** on MeVIS (val) (54.8 vs. 50.0).

---

> ### Author Response · Authors · 2025-11-19
> **Reponse to Reviewer CUfj (Q2)**
>
> **Q2:** Similarity to Prior Grounding Work: The approach of using the [FIND] token to localize keyframes is highly similar to the established "streaming EOS prediction" mechanism found in related work, such as VideoLLM-Online, which diminishes the technical novelty of this specific component.
>
>
> **A2:** We thank the reviewer for this comment and would like to clarify that our **[FIND] token mechanism** is fundamentally different from the streaming **[EOS] token** mentioned.
>
> * **Different Tasks:** MomentSeg is an **offline processing framework** designed for RefVOS. In contrast, the [EOS] (End of Sequence) token in models such as VideoLLM-Online is primarily used for managing streaming dialogues or skipping processing steps.
>
> * **Different Roles:** The [FIND] token serves as a semantic query to identify frames most relevant to the input text, enabling the localization of key video segments for RefVOS segmentation. By comparison, the [EOS] token in VideoLLM-Online signals the transition to the next frame, helping the model skip redundant computations for irrelevant frames.
>
> * **[FIND] Operation Mechanism:** The [FIND] token acts as a semantic query embedding. It computes a feature similarity against all frame-level pooled embeddings. For example, for N input frames, it generates N scores, with each score representing the relevance of that specific frame to the input sentence. This similarity distribution is the core mechanism that enables our model to perform TSG and, subsequently, Moment-Centric Sampling.
>
> While online RefVOS represents an important research direction, it is beyond the scope of our current work. We have added a discussion of the idea of VideoLLM-Online [1] for future work in the **Section F.6** of the revised manuscript. For more details on [FIND], please refer to **A1 for Reviewer pifY**.
>
> [1] VideoLLM-online: Online Video Large Language Model for Streaming Video, CVPR 2024

---

> ### Author Response · Authors · 2025-11-19
> **Reponse to Reviewer CUfj (Q3)**
>
> **Q3:** Limited Applicability to Online Scenarios: Key components of the proposed method, including Moment-Centric Sampling (MCS) and Bidirectional Anchor-updated Propagation (BAP), fundamentally require access to the entire video's context, severely limiting their applicability to real-world online or streaming environments.
>
> **A3:** We agree with the reviewer that our current model is designed for **offline** processing. Key components like Moment-Centric Sampling (MCS) and Bidirectional Anchor-updated Propagation (BAP) do require access to the full video context and are not applicable to online or streaming scenarios. However, this does not diminish the significance of offline research, as the majority of existing RefVOS methods are also developed in an offline setting, reflecting the practical usage scenarios.
>
> We would like to clarify that many important real-world applications, such as `e-commerce` and `video editing`, require this high-performance offline capability. From an information perspective, offline (bidirectional) processing, by having access to the entire video, can generally achieve higher accuracy than causal (online) models. Many temporal tasks, such as object tracking and video object detection, have both offline and online settings.
>
> The focus of this paper was to maximize performance in the offline setting. We agree that real-world online and streaming RefVOS is a very valuable and important research direction. We thank the reviewer for this suggestion and have added this as a limitation and area for future work in the revised manuscript.

---

> ### Author Response · Authors · 2025-11-19
> **Reponse to Reviewer CUfj (Q4)**
>
> **Q4:** Novelty of Bidirectional Propagation: While effective within this framework, the concept of bidirectional propagation for ensuring temporal consistency is not a novel technique in the broader Video Object Segmentation (VOS) literature and represents mainly an adaptation to the SAM model rather than a core innovation.
>
> **A4:** We thank the reviewer for this comment. We would like to respectfully clarify the purpose of our Bidirectional Anchor-updated Propagation (BAP).
>
> The primary goal of BAP is **not** to ensure temporal consistency. As the reviewer notes, SAM2 is a video model, and its temporal consistency largely stems from its built-in memory bank mechanism, which models temporal features from previous frames. While temporal consistency is crucial in VOS, and many methods (e.g., VITA, which implicitly construct consistency by using a single query to represent the entire video instance, or the DVIS series, which introduces a tracker and uses Hungarian matching to organize query consistency) address it, our BAP is designed to solve two different, specific problems:
>
> * **Why Bidirectional Propagation?** This is to handle **late-appearing objects**. For example, in a 100-frame video, if the target only appears from frame 50-60, a standard forward propagation (starting from frame 1) might fail entirely if it doesn't happen to sample the object's appearance at frame 50. BAP solves this by **initializing propagation from the central keyframe** (e.g., frame 55) identified by MCS, where the target is guaranteed to exist. It then propagates both forwards and backwards. This ensures robust initialization without increasing the total number of frames processed or the computational cost.
> * **Why Anchor-Updating?** This is for **error correction**. If the initial mask on the keyframe is incorrect, this error would propagate throughout the entire video, as it is never corrected. The "updating" mechanism in BAP uses the *other* keyframes sampled by MCS as anchor points to **update and correct the mask** during propagation. As our ablation in **Table 9** demonstrates, this updating step is critical and provides a significant performance boost.

---

> ### Author Response · Authors · 2025-11-19
> **Reponse to Reviewer CUfj (Q5)**
>
> **Q5:** Missing Robustness Analysis: The paper lacks a deep-dive analysis on the method's robustness, particularly how BAP performs when anchor frames contain significant noise or the model encounters high degrees of occlusion, which are common challenges in propagation-based models.
>
> **A5:** We thank the reviewer for raising this important point regarding the robustness of BAP. To address this concern, we have performed a comprehensive robustness analysis on the MeViS ($val^u$) dataset and incorporated the results into the revised manuscript (**Table 13** and **Sec. 4.3** *Analysis of Robustness*).
>
> Our analysis focuses on two major factors that determine the robustness of BAP: **the quality of the initial mask** and **the effectiveness of the updating mechanism**.
>
> **Initial Mask Quality.**
> We evaluate two metrics—*Initial Shot Ratio* (the percentage of initial frames that contain the referred target) and *Initial mIoU* (the segmentation quality on the initial frame). MomentSeg achieves an Initial Shot Ratio exceeding **95%**, indicating that it reliably selects starting frames where the object is visible. Moreover, compared with Sa2VA, the Initial mIoU improves by **9.3%**, demonstrating that BAP provides high-quality initial masks, which is crucial for robust propagation.
>
> **Effectiveness of Updating.**
> To assess the correction ability of BAP, we compare the mIoU of anchor frames with and without updating. Anchor updating yields a **2.6%** improvement in mIoU, confirming that BAP can effectively refine and correct masks when propagation errors occur. Additionally, we report the *Correction Ratio*, defined as the proportion of frames with IoU < 0.3 that are corrected to IoU > 0.7 through updating. BAP corrects approximately **7%** of such error cases, highlighting its robustness against challenging scenarios such as heavy occlusion and noise in anchor frames.
>
> **Table: Robustness evaluation of BAP on the MeViS ($val^u$) dataset.** Initial Shot Ratio denotes the percentage of initial frames containing the referred target. Initial mIoU measures the segmentation quality on the initial frame. Anchor mIoU compares performance without and with anchor updating. Correction Ratio indicates the proportion of frames with IoU \<0.3 that are corrected to IoU \>0.7 via updating.
>
> | Method                        | Initial Shot Ratio (%) | Initial mIoU (%) | Anchor mIoU (w/o → w/ Updating) (%) | Correction Ratio (%) |
> | :---------------------------- | :--------------------: | :--------------: | :----------------------------------: | :------------------: |
> | Sa2VA-8B                      |          88.7          |       54.0       |                  --                  |          --          |
> | **MomentSeg-7B (Ours)** |      96.6 (↑7.9)      |   63.3 (↑9.3)   |         61.0 → 63.6 (↑2.6)         |         7.1         |

---

> ### Comment · Area_Chair_1DmS · 2025-11-25
>
> Dear Reviewer CUfj,
>
> The authors have responded to your reviews. Please review and provide your feedback and responses.
>
> Best,
>
> Your AC

---

> ### Author Response · Authors · 2025-11-27
> **Comment by Authors**
>
> Dear Reviewer,
>
> Thank you again for your time and valuable suggestions.
>
> As the rebuttal period is approaching its end with **less than one week remaining**, we wanted to kindly check if there are any remaining concerns or questions that we could help clarify.
>
> If you have **any additional points or technical questions**, please feel free to raise them — we would try our best to provide further explanation or experiments to support a thorough evaluation.
>
> We sincerely appreciate your insights and look forward to your reply.
>
> Best regards,
> Authors of Paper 1310

---

### Official Review · Reviewer_pifY · 2025-10-31

**Soundness:** 3
**Presentation:** 3
**Contribution:** 3
**Rating:** 6
**Confidence:** 3

**Summary:**

This work proposes a unified framework that jointly optimizes temporal sentence grounding and segmentation. A novel [FIND] token enables key moment identification without external timestamps, while a Moment-Centric Sampling (MCS) strategy balances dense and sparse frame sampling for efficiency and context preservation. Additionally, a Bidirectional Anchor-updated Propagation (BAP) mechanism enhances tracking stability by adaptively updating masks during inference.

**Strengths:**

1.	The paper addresses a valuable research question: Is keyframe selection necessary?
2.	The experiments cover a wide range of tasks and datasets, with comprehensive baseline comparisons.
3.	The ablation studies are well-designed and clearly demonstrate the contribution of each component.
4.	The proposed method achieves competitive performance.

**Weaknesses:**

1.	The FIND token plays a critical role in the proposed approach. However, the paper does not clearly explain how this token is supervised or optimized. Moreover, the semantic role of the FIND token within the model (e.g., whether it acts as a query, indicator, or moment aggregator) remains unclear.
2.	The Moment-Centric Sampling design appears rather complex, but the authors lacks justification for its necessity and detailed analysis of why this design is better.
3.	The overall framework is complicated with many hyperparameters, which increases the tuning cost.
4.	The inference efficiency is not adequately discussed. Considering the method includes multiple tokens and multiple steps, the inference cost may be relatively high.
5.	Figures and writing quality could be further polished for clarity and readability.

**Questions:**

See weaknesses.

---

> ### Author Response · Authors · 2025-11-19
> **Response to Reviewer pifY (Q1)**
>
> We sincerely thank the reviewer for your time and constructive feedback. We have carefully considered all suggestions and have addressed them point-by-point below. We have also included additional experiments and analyses where appropriate, and we hope these revisions and clarifications resolve the reviewer's concerns.
>
> **Q1:** The FIND token plays a critical role in the proposed approach. However, the paper does not clearly explain how this token is supervised or optimized. Moreover, the semantic role of the FIND token within the model (e.g., whether it acts as a query, indicator, or moment aggregator) remains unclear.
>
> **A1:** We thank the reviewer for the question regarding the [FIND] token. We summarize the role of [FIND] and its supervision process as follows:
>
> * **Semantic Role:** The [FIND] token functions as a **semantic query embedding**. It implicitly learns to represent the semantic information of the input *sentence*. Its role is to "**query**" the video frames to identify the most relevant temporal moment.
> * **Supervision and Optimization:** The [FIND] token is optimized through the TSG task. We first compute the feature similarity between the output embedding of the [FIND] token and each frame-level pooled feature, producing a set of relevance scores (e.g., N scores for N frames) that measure how relevant each frame is to the sentence query. We then supervise these relevance scores using a contrastive learning objective. Through this optimization, the model learns to treat the [FIND] token as an effective query for scoring and identifying text-relevant frames. This process is illustrated in the lower-left part of **Fig. 3** and **its caption**, and the computation of the supervision term involving the [FIND] token is formalized in **Equation 5** as $\mathcal{L}_{\text{find}}$. We will clarify this further in updated versions.

---

> ### Author Response · Authors · 2025-11-19
> **Response to Reviewer pifY (Q2)**
>
> **Q2** The Moment-Centric Sampling design appears rather complex, but the authors lacks justification for its necessity and detailed analysis of why this design is better.
>
> **A2:** We thank the reviewer for this insightful comment. We apologize if the description of Moment-Centric Sampling (MCS) made it appear overly complex. We would like to clarify its efficiency, necessity, and the analysis supporting its design.
>
> * **Low Complexity and Cost:** We would first like to clarify that MCS is computationally lightweight. The core logic (detailed in **Algorithm 1**) involves simple CPU operations: (1) *Compute partitioned similarity mass*, (2) *Allocate partition sampling budget*, and (3) *Perform stratified sampling*. Our analysis shows its average latency is only **~5ms** per video, accounting for **less than 0.1%** of the total average inference latency (~7s).
> * **Necessity of the Design:** The primary goal of MCS is to provide the subsequent RefVOS task with a *small but highly relevant* set of keyframes. We discussed the motivation for this design in the main paper (Lines 127-132). Simpler methods, like the 'FirstK' sampling used in Sa2VA, often fail because the relevant event may occur later in the video. MCS solves this by **densely sampling frames around the specific "moment"** identified as most text-relevant, while sampling sparsely elsewhere. This ensures the model receives the most critical visual information efficiently. We previously analyzed its effectiveness in **Section 4.3 ("Effectiveness of Moment-Centric Sampling")**. To provide a clearer and more concise description of MCS, we have revised the description of **Algorithm 1** in **Section 3.2.2**.
> * **Analysis of Effectiveness:**: **From Quantitative:** As shown in **Table 8**, our MCS strategy substantially outperforms the 'FirstK' sampling baseline, achieving a **+3.0%** gain on MeViS ($val^u$) and a **+4.9%** gain on ReasonVOS. **From Qualitative:** We provide visual comparisons in **Figure 8** to illustrate how MCS helps capture the correct event compared to Sa2VA. Furthermore, we have added a detailed discussion in the **Appendix (Section F.1)** analyzing specific cases where MCS is advantageous and explaining the role of this sampling method.

---

> ### Author Response · Authors · 2025-11-19
> **Response to Reviewer pifY (Q3)**
>
> **Q3:** The overall framework is complicated with many hyperparameters
>
> **A3:**
> For the loss function, the key hyperparameters are the temperature factor $\tau$ and the positive sample weight $\lambda_p$. The temperature $\tau$ is a standard parameter for scaling cosine similarity, and we directly adopt the default value from CLIP. The positive weight $\lambda_p$ is used to address the significant imbalance between positive (event) and negative (background) frames in TSG datasets. We found $\lambda_p=2$ yields good results, and we include an ablation study (reproduced below) to show its impact and justify this choice.
>
> | $\lambda_p$ |     R@0.3     |     R@0.5     |     R@0.7     |      mIoU      |
> | :-----------: | :------------: | :------------: | :------------: | :------------: |
> |      0.5      |      74.8      |      49.2      |      18.9      |      47.8      |
> |      1.0      |      72.5      |      49.7      |      22.6      |      48.3      |
> | **2.0** | **77.2** | **51.4** | **24.9** | **49.9** |
> |      3.0      |      75.9      |      50.6      |      23.3      |      49.5      |
>
> The other main hyperparameter is the threshold $\theta$ used in TSG post-processing. It is important to note that $\theta$ **only affects inference** and is not involved in training, thus not increasing the tuning cost. Its role is analogous to the confidence threshold in object detection, used to filter which temporal moments are predicted. This provides necessary flexibility, as users can dynamically adjust $\theta$ to meet their desired precision/recall trade-off. We provided a detailed analysis of $\theta$'s impact in **Table 21 of the Appendix**. The analysis of frame number $N_f$ is provided in **Table 20 of Appendix**. The ablation studies of the temporal frame number is provided in **Figure 6,7 in Appendix**.

---

> ### Author Response · Authors · 2025-11-19
> **Response to Reviewer pifY (Q4,Q5)**
>
> **Q4:** The inference efficiency is not adequately discussed. Considering the method includes multiple tokens and multiple steps, the inference cost may be relatively high.
>
> **A4:** We thank the reviewer for this helpful suggestion regarding inference efficiency. We have analyzed this part in detail in our Global Comment (please see **G2**).
>
> In summary, MomentSeg achieves a 10-point J&F improvement over Sa2VA in end-to-end inference on MeViS ($val$), with only a 3% decrease in inference speed. More analysis can be found in **G2** or **Table 12** in the revised paper.
>
> **Q5:** Figures and writing quality could be further polished for clarity and readability.
>
> **A5:** We thank the reviewer for this suggestion. We have revised the manuscript to improve the clarity of the descriptions and figures in the updated version.

---

> ### Comment · Area_Chair_1DmS · 2025-11-25
>
> Dear Reviewer pifY,
>
> The authors have responded to your reviews. Please review and provide your feedback and responses.
>
> Best,
>
> Your AC

---

> ### Author Response · Authors · 2025-11-27
> **Comment by authors**
>
> Dear Reviewer,
>
> Thank you again for your time and valuable suggestions.
>
> As the rebuttal period is approaching its end with **less than one week remaining**, we wanted to kindly check if there are any remaining concerns or questions that we could help clarify.
>
> If you have **any additional points or technical questions**, please feel free to raise them — we would try our best to provide further explanation or experiments to support a thorough evaluation.
>
> We sincerely appreciate your insights and look forward to your reply.
>
> Best regards,
> Authors of Paper 1310

---

### Author Response · Authors · 2025-11-18
**Global Comment**

Here we provide some global comments that will be referenced in subsequent one-on-one replies.

---

> ### Author Response · Authors · 2025-11-18
> **G1 Core contributions of MomentSeg**
>
> 1. In the RefVOS domain, previous methods often relied on external pre-trained keyframe models to improve RefVOS performance. MomentSeg **inherently** possesses TSG capabilities and no longer depends on external models (as concluded in  "*External Keyframe Model*" in Table 1).
> 2. MomentSeg utilizes a **[FIND] token** to determine the similarity distribution with temporal frames, which obviates the need for extra timestamp encoding typically required in TSG tasks (as noted in "*Timestamps Encoding in Temporal Sentence Grounding*" in Related Work).
> 3. The design of **Moment-Centric Sampling (MCS)**, which operates with **minimal overhead** (averaging 5ms per video on a CPU), provides **dense** sampling for text-relevant key regions and **sparse** sampling for other areas. This preserves both critical motion information and global context.
> 4. The design of the  **Bidirectional Anchor-updated Propagation (BAP)** assists the SAM2 model by utilizing the key moment as an **initialization** to generate a high-quality mask. It then performs mask propagation both toward the video's start and end, and dynamically **updates** the mask during this propagation to mitigate the accumulation of errors (see Table 13 for detailed analysis).
> 5. Our MomentSeg-7B achieves significant improvements over Sa2VA-8B, including **+10.2% on MeViS** and **+7.5% on ReVOS**. It also supports multiple tasks, including TSG, RefSeg, VideoQA, and ImageQA，The main paper and appendix present metrics on **20+ datasets** to demonstrate MomentSeg's superior performance and robustness across these various tasks.

---

> ### Author Response · Authors · 2025-11-18
> **G2 Inference Latency Analysis**
>
> We have conducted a more detailed analysis of the latency, which we hope helps illustrate our method's practical efficiency.
>
> 1. **Latency Decomposition:** To provide a clear breakdown, we analyzed the end-to-end latency by separating it into three components: **Preprocess**, **MLLM** , and **SAM2**.
> 2. **MLLM Latency Analysis:** Our analysis suggests that the `Preprocess` and `SAM2` stages account for the majority of the total runtime. The MLLM component, which performs two forward passes (one for TSG, one for RefVOS), contributes a relatively smaller and stable portion. For a typical video, the MLLM processes a limited number of frames (approx. 2–4k tokens).
> 3. **Quantitative Breakdown:** As shown in Table below, the MLLM component accounts for approximately **20%** of the total latency (~8% for TSG and ~12% for RefVOS). We noted that latency variations across datasets are primarily driven by differences in video length and resolution, which impact the `Preprocess` and `SAM2` stages, while the MLLM latency remains relatively constant.
> 4. **Efficiency and Performance:** We also compared our method to the Sa2VA-8B baseline. We found that MomentSeg-7B introduced only a minor increase in the total inference latency, approximately **3%**, while also achieving performance gains (**+10.2% on MeVIS** and **+7.5% on ReVOS**). We believe this comparison highlights a favorable balance between accuracy and efficiency.
>
> We have included this detailed breakdown and discussion in the revised manuscript in **Section 4.3, under "Analysis of Inference Latency,"** to better address this point.
>
> **Comparison of inference efficiency and performance.** The “Latency” denotes the end-to-end inference time per video, measured on a single NVIDIA A100 GPU (40 GB).
>
> | Method | Ref-Youtube-VOS (J&F) | Ref-Youtube-VOS (Latency (s)) | MeVIS (val) (J&F) | MeVIS (val) (Latency (s)) | ReVOS (J&F) | ReVOS (Latency (s)) |
> | :--- | :---: | :---: | :---: | :---: | :---: | :---: |
> | Sa2VA-8B | 70.7 | 6.8 | 46.9 | 7.5 | 57.6 | 6.3 |
> | **MomentSeg-7B (Ours)** | 72.3(↑1.6) | 7.0(↓3%) | 57.1(↑10.2) | 7.7(↓3%) | 65.1(↑7.5) | 6.4(↓2%) |
> | ↳ Preprocess | - | 1.8 (26%) | - | 3.3 (43%) | - | 2.4 (38%) |
> | ↳ MLLM | - | 1.3 (19%) | - | 1.4 (18%) | - | 1.3 (20%) |
> | **&emsp;**↳ TSG | - | 0.5 (7%) | - | 0.6 (8%) | - | 0.5 (8%) |
> | **&emsp;**↳ RefVOS | - | 0.8 (12%) | - | 0.8 (10%) | - | 0.8 (12%) |
> | ↳ SAM2 | - | 3.9 (55%) | - | 3.0 (39%) | - | 2.7 (42%) |

---

### Author Response · Authors · 2025-11-18
**Rebuttal Summary and Revisions**

Dear AC and Reviewers,

We sincerely thank you for your thoughtful reviews and constructive feedback, which have greatly strengthened our submission. All corresponding revisions have been incorporated into the updated manuscript (highlighted in blue). Should any part of our response remain unclear, we would be glad to provide further clarification.

We are encouraged by the positive remarks from the reviewers:

* **pifY：** Valuable Analysis, comprehensive experiments, well-designed ablations, and competitive performance.
* **CUfj：** Clear motivation, comprehensive strong results, and a clear, well-structured methodology
* **62dL：** Extensive evaluations with consistent improvements, well-motivated design, thorough ablations, strong performance and clear presentation with rich supplementary material.
* **zWDN：** Conceptually sound formulation with rigorous methodology, clear motivation and systematic, reproducible ablations.

Following your valuable suggestions, we have made the following improvements:

- Added a detailed **Inference Latency** analysis in **Table 12** of the main paper.
- Added a **robustness evaluation of BAP** in **Table 13**.
- Added evaluation results on the **Ref-SAV dataset** in **Table 5**, showing that MomentSeg-7B surpasses the SoTA model UniPixel-7B (NeurIPS 2025) by **11.8%** and the baseline Sa2VA-8B by **39.5%** in the J&F metric.
- Added **Section F.6**, discussing additional directions such as **streaming video segmentation** (e.g., [EOS] design in VideoLLM-Online), adaptation to tasks like **TAS** (GTEA, Breakfast), and related models that merit further exploration (MS-TCN++, ASFormer).
- Revised **Section 4.3** (“*Effectiveness of Moment-Centric Sampling*”) to improve clarity and strengthen the analysis of MCS advantages, including a more explicit discussion of its complexity.
- Rewrote **Algorithm 1 (Section 3.2.2)** to present Moment-Centric Sampling as a systematic three-step procedure: (1) Compute partitioned similarity mass; (2) Allocate partition sampling budget; (3) Perform stratified sampling.

We sincerely appreciate your time and engagement throughout the discussion period. Your feedback has meaningfully improved the quality and clarity of our work. We warmly welcome any additional comments or suggestions.

Sincerely,

Authors of Paper 1310

---

### Author Response · Authors · 2025-11-28

Dear Reviewers,

Thank you again for your time and valuable suggestions.

As the rebuttal stage is approaching its end, could we kindly ask for your feedback or comments on the efforts we have made during the rebuttal? We would be very grateful for your reply.

Best regards,
Authors of Paper 1310

---

### Meta-Review · Area_Chair_gXtZ · 2025-12-30

**Summary:**

The common concern raised by multiple reviewers is that the overall framework is somehow complicated and involves many components and hyperparameters, which increases the tuning cost and also lacks a deep-dive analysis. After reading the paper myself, I share the similar impression: the framework feels complex yet not particularly novel, such as eliminating external models but leveraging a [FIND] token for the module of keyframe selection. The novelty of the whole paper appears fairly limited. In addition, the paper does not adequately discuss inference cost and efficiency: at least three reviewers noted that it does not clearly report inference time or computational overhead (the authors provided this in their revision, showing a slight increase in inference time).

**Reviewer Concerns:**

Reviewer zWDN's concern about the moment definition and limited temporal boundary precision appears to have been addressed in the rebuttal.

However, the major concerns remain outstanding: the overall framework is overly complex with many hyperparameters, lacks sufficient justification for why such a design is necessary, and provides limited analysis explaining why this design is better. Questions about the limited applicability and the novelty of the proposed components were raised by almost all reviewers.

**Reviewer Scores:**

Reviewer zWDN responded positively to the rebuttal, while he had already given a positive score initially. The other reviewers are unlikely to change their scores based on the current discussion.

---

### Decision · Program_Chairs · 2026-01-26

Reject